COMMUNICATIONS

# Deficiency in endocannabinoid synthase *DAGLB* contributes to early onset Parkinsonism and murine nigral dopaminergic neuron dysfunction

Zhenhua Liu[1,2,23], Nannan Yang[1,2,23], Jie Dong[1,3,23], Wotu Tian[1,4], Lisa Chang[1], Jinghong Ma[5], Jifeng Guo [2], Jieqiong Tan[6], Ao Dong [7,8,9], Kaikai He [7,8], Jingheng Zhou [10], Resat Cinar [11], Junbing Wu[1], Armando G. Salinas [12], Lixin Sun[1], Mantosh Kumar[1], Breanna T. Sullivan[1], Braden B. Oldham [1], Vanessa Pitz [13], Mary B. Makarious [13], Jinhui Ding[14], Justin Kung[1], Chengsong Xie[1], Sarah L. Hawes[1], Lupeng Wang[1], Tao Wang[15], Piu Chan[5], Zhuohua Zhang[6,16], Weidong Le[3,17], Shengdi Chen[4], David M. Lovinger[12], Cornelis Blauwendraat [18], Andrew B. Singleton[13,19], Guohong Cui [10], Yulong Li [7,8,9,20], Huaibin Cai [1,24✉] & Beisha Tang [2,6,21,22,24✉]

Endocannabinoid (eCB), **2**-arachidonoyl-glycerol (2-AG), the most abundant eCB in the brain, regulates diverse neural functions. Here we linked multiple homozygous loss-of-function mutations in 2-AG synthase diacylglycerol lipase β (*DAGLB*) to an early onset autosomal recessive Parkinsonism. DAGLB is the main 2-AG synthase in human and mouse *substantia nigra* (SN) dopaminergic neurons (DANs). In mice, the SN 2-AG levels were markedly correlated with motor performance during locomotor skill acquisition. Genetic knockdown of *Daglb* in nigral DANs substantially reduced SN 2-AG levels and impaired locomotor skill learning, particularly the across-session learning. Conversely, pharmacological inhibition of 2-AG degradation increased nigral 2-AG levels, DAN activity and dopamine release and rescued the locomotor skill learning deficits. Together, we demonstrate that *DAGLB*-deficiency contributes to the pathogenesis of Parkinsonism, reveal the importance of DAGLB-mediated 2-AG biosynthesis in nigral DANs in regulating neuronal activity and dopamine release, and suggest potential benefits of 2-AG augmentation in alleviating Parkinsonism.

---

A full list of author affiliations appears at the end of the paper.

Parkinson's disease (PD) is clinically manifested with both motor and non-motor symptoms[1]. A preferential degeneration of midbrain dopaminergic neurons (DANs) in the ventral *substantia nigra pars compacta* (SNc) and the resulting impairments of dopamine transmission in basal ganglia are broadly responsible for the motor symptoms, which include bradykinesia, resting tremor, rigidity, and motor skill learning deficits[2,3]. Both genetic and environmental factors contribute to the etiopathogenesis of PD or Parkinsonism. The identification of monogenetic mutations responsible for various familial forms of PD or Parkinsonism provides molecular clues in understanding the pathophysiological mechanisms of the disease[4]. To date, more than 20 genes have been linked to the familial forms of PD or Parkinsonism, including both autosomal dominant and recessive mutations[5,6]. However, a significant proportion of familial PD or Parkinsonism cases are still genetically unknown. Identifying those unknown genetic factors may uncover additional signaling pathways in regulating nigral DAN activity and the pathogenesis of PD or Parkinsonism.

The nigral DANs are essential in regulating the vigor of movement[7] and motor learning[8]. The activity of nigral DANs and dopamine release can be dynamically regulated by diverse presynaptic inputs, of which the direct pathway striatal spiny projection neurons (dSPNs) in the dorsal striatum provide the major inhibitory inputs[8–12]. High levels of cannabinoid receptor 1 (CB1) are present in the axon terminals of dSPNs[13,14], which may respond to the endocannabinoid (eCB) released from the nigral DANs, including 2-arachidonoyl-glycerol (2-AG) and N-arachidonoylethanolamine (AEA). The eCBs act as neuromodulators, retrogradely suppressing presynaptic neurotransmitter release through the G protein-coupled CB1 receptors, and regulating a variety of physiological processes, such as motor learning, stress response, and memory[15–19]. The midbrain DANs can produce and release eCBs from their soma and dendrites[20]. Both diacylglycerol lipase α (DAGLA) and its homolog DAGLB mediate the biosynthesis of 2-AG, the most abundant eCB in the brain[21]. While DAGLA catalyzes most of the 2-AG production in the brain[22–24], the main 2-AG synthase in nigral DANs remains to be determined. Confounding upregulation and downregulation of eCBs and associated receptors have been observed in the basal ganglia of patients with PD and related animal models[18,25,26]. However, it is unclear whether the alterations of eCB signaling contribute to the etiopathogenesis of disease or merely reflect the compensatory responses during the progression of disease. Understanding how the eCB system regulates dopamine transmission in motor control and learning may provide additional insights into the pathogenic mechanisms of PD or Parkinsonism.

To support the involvement of eCB signaling in regulating nigral DAN activity and pathogenesis of Parkinsonism, here we provided genetic evidence to demonstrate that deficiency in 2-AG synthase *DAGLB* contributes to the etiopathogenesis of Parkinsonism. We then reveal a nigral DAN-specific pathogenic mechanism of *DAGLB* dysfunction in locomotor skill learning. Finally, we show that pharmacological augmentation of 2-AG levels may serve as a potential therapeutic treatment for Parkinsonism.

## Results

### Identification of *DAGLB* mutations in patients with early onset autosomal recessive Parkinsonism.

Previously, we recruited a large cohort of patients with autosomal recessive Parkinsonism (ARPD) and sporadic early onset Parkinsonism (EOPD) in China and identified pathogenic variants of known PD genes using exon dosage analysis and whole-exome sequencing (WES)[27]. To discover unknown causal genetic mutations in the remaining 171

ARPD families after excluding those with known PD genes, we first studied one consanguineous family (Family 1, AR-003) with two siblings affected by EOPD (Fig. 1a). Genome-wide single nucleotide polymorphism (SNP) analysis and homozygosity mapping of the affected individuals revealed five regions of homozygosity shared by the affected sisters (II-3 and II-4) as the candidate causative gene regions (Supplementary Table 1, Supplementary Fig. 1). Assuming recessive mode of inheritance, we then analyzed the WES data from those two affected siblings and searched for shared homozygous mutations. Consequently, we identified one homozygous splice-site mutation (c.1821-2A>G) residing in intron 14 of *DAGLB* confirmed by Sanger sequencing and segregated with disease in this family (Fig. 1a, Supplementary Tables 2 and 3, Supplementary Fig. 2). The "c.1821-2A>G" mutation was predicted in silico to disrupt the donor splice site of exon 15 and confirmed by reverse-transcription PCR analysis (Supplementary Fig. 3). Next, we analyzed the *DAGLB* gene for homozygous or compound heterozygous mutations by mining the WES data from an additional 1,741 unrelated PD probands, including 170 ARPD and 1,571 sporadic EOPD cases without known PD-related genetic mutations. Accordingly, we identified one homozygous missense mutation [c.1088A>G (p.D363G)] in Family 2 (AR-005) and one homozygous frameshift mutation [c.469dupC (p.L158Sfs*17)] in Family 4 (Fig. 1a, Supplementary

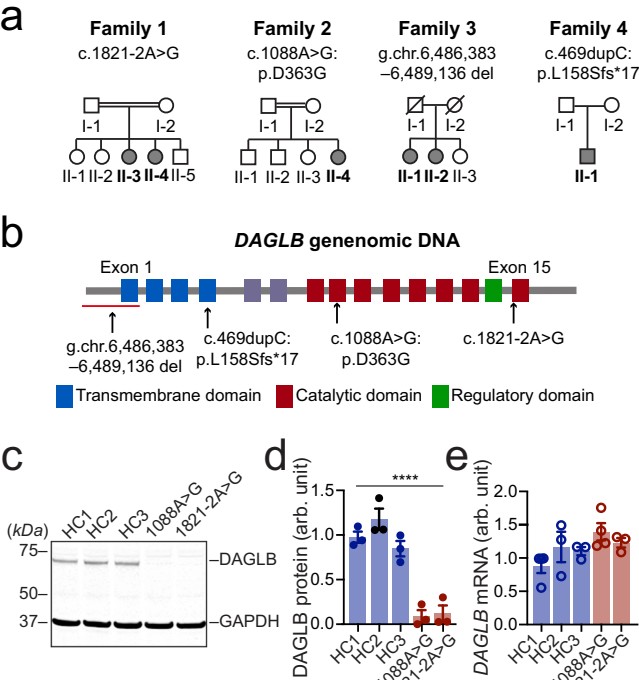

**Fig. 1 Identification of homozygous *DAGLB* mutations in affected families. a** Pedigrees are shown for the four affected families. A double bar represents parental consanguinity. Slash indicates deceased individuals. Males are represented by squares, females by circles, and affected individuals by shading. **b** Schematic view of *DAGLB* gene structure and encoded protein domains. The four transmembrane segments are shown in blue, and the catalytic domain in maroon. Within the catalytic domain, a regulatory loop is colored in green. Variant sites are indicated by arrows. **c–e** Representative western blot (**c**) and quantification of *DAGLB* protein (**d**) and mRNA (**e**) levels in fibroblasts derived from Family 1 II-3, Family 2 II-4 and three age-matched HCs. $N = 3$–4 independent fibroblast cultures. Data were normalized to glyceraldehyde 3-phosphate dehydrogenase (GAPDH) protein or mRNA levels as appropriate and represent mean ± SEM. One-way ANOVA, $F = 33.09$, ****$p < 0.0001$.

Tables 2 and 3, Supplementary Fig. 2). Genome-wide SNP array genotyping also showed homozygosity present in the affected cases from Families 2 and 4, which include the *DAGLB* gene (Supplementary Fig. 1). Finally, we performed copy-number variation analysis of the WES data and identified one more homozygous deletion (g.ch7:6,486,383-6,489,136del), which contains the entire exon 1 and 5'-untranslated region of *DAGLB* gene in another family with two affected siblings (Family 3, AR-075) and was validated through Oxford Nanopore long-read sequencing and Sanger sequencing (Fig. 1a, Supplementary Figs. 4 and 5). These *DAGLB* mutations are absent from or present in the heterozygous state in available unaffected family members and healthy control participants, strongly supporting the pathogenicity of homozygous *DAGLB* mutations in EOPD. Those six affected individuals had early disease onset (≤40 years old) and displayed typical Parkinsonism and good levodopa response (Supplementary Table 4). Positron emission tomography (PET) revealed impaired dopamine transmission in the striatum (Supplementary Fig. 6). Together, although exceedingly rare, we linked four different homozygous *DAGLB* mutations to six affected ARPD/EOPD individuals from four different families of Chinese descent.

In addition to the Chinese population, we conducted a preliminary search for any potential *DAGLB* pathogenic variants in the Accelerating Medicines Partnership - Parkinson's disease (AMP-PD) dataset, which includes 2,556 controls and 1451 PD cases from unrelated European descent. Using a minor allele frequency (MAF) < 0.05, we identified 30 missense variants and there were no loss-of-function mutations (splicing, stop, frameshift) in the *DAGLB* gene. Following the proposed mechanism of recessive disease, we found 12 out of 1451 PD cases were recessive or (potentially) compound heterozygous, which results in a frequency of 0.008. Additionally, 28 out of 2,556 controls were recessive or (potentially) compound heterozygous which results in a frequency of 0.01. We also used SKAT-O, CMC Wald, and CMC burden testing, which resulted in non-significant $p$-values for all missense variants as defined by ANNOVAR: 1, 0.94, and 0.94, respectively. This analysis suggests that *DAGLB* is not linked to sporadic late-onset PD cases of European descent.

**The Parkinsonism-related mutations disrupt the formation and stability of *DAGLB* protein.** *DAGLB* encodes a 672-amino acid protein containing four transmembrane domains and one catalytic domain that mediates the biosynthesis of 2-AG[24]. The "g.ch7:6,486,383-6,489136del" and "c.469dupC" mutations apparently disrupt the translation of the catalytic domain (Fig. 1b), resulting in loss-of-function of DAGLB. By contrast, the "c.1821-2A>G" mutation truncates part of the catalytic domain, while the "c.1088A>G" mutation replaces a conserved aspartate (D) residue (D363) with glycine (G) in the catalytic domain (Fig. 1b, Supplementary Fig. 7a). D363 resides in a β-strand within an 8-strand β-sheet structure and the β-sheet is the core of an α-β-α tertiary structure predicted by AlphaFold[28] (Supplementary Fig. 7b). The D363G missense mutation replaces an acidic sidechain with a non-polar sidechain, which may inevitably change the electrostatic and Van der Waals interactions in the pocket and affect the normal folding of the catalytic domain. To investigate whether the splicing and missense mutations affect the expression of DAGLB protein, we examined the expression of DAGLB protein and *DAGLB* mRNA in primary fibroblasts derived from patients carrying the mutations. Compared to the healthy controls (HC), DAGLB protein was barely detectable by western blot in the patients' samples [one-way ANOVA, $F(4, 10)$ = 33.1, $p < 0.0001$] (Fig. 1c, d). In contrast, the *DAGLB* mRNA expression is comparable between HC and patients' samples

[one-way ANOVA, $F(4, 10)$ = 2.3, $p = 0.12$] (Fig. 1e), suggesting that the mutations affect the stability of DAGLB protein. Indeed, treatment with proteasome inhibitor MG132 increased the levels of DAGLB protein in both HC and patients' samples (Supplementary Fig. 8a, b). Additionally, the mutations did not affect the expression of DAGLA protein in patients' samples (Supplementary Fig. 8c, d). Therefore, all four Parkinsonism-related mutations disrupt the formation or stability of DAGLB protein, suggesting that the impairment of *DAGLB*-mediated 2-AG signaling may contribute to the etiopathogenesis of Parkinsonism.

**DAGLB is the main 2-AG synthase expressed in nigral DANs.** While our human genetic studies linked a deficiency in *DAGLB* to Parkinsonism (Fig. 1), *DAGLA* is the main 2-AG synthesis in the CNS and account for 80% production of 2-AG in the mouse brains[22,23]. How does the *DAGLB*-deficiency contribute to Parkinsonism and nigral DAN dysfunction? To address this concern, a previous whole-genome RNA-sequencing study[29] revealed 10-fold more abundance of *DAGLB* than *DAGLA* mRNA levels in laser capture microdissection (LCM)-isolated human SN DANs (Fig. 2a, unpaired $t$ test, $p < 0.0001$, plotted from GSE76514). We then performed RNA-sequencing of LCM-isolated mouse nigral DANs and found that the expression of *Daglb* mRNA is 2-fold higher than *Dagla* mRNA (unpaired $t$ test, $p < 0.0001$, plotted from PRJNA775656) (Fig. 2b). By contrast, *Dagla* mRNA was more enriched in striatal neurons than *Daglb* mRNA (unpaired $t$ test, $p < 0.0001$, plotted from PRJNA6124778 associated with a previous publication[30]) (Fig. 2c). To determine the cellular location of DAGLB protein in nigral DANs, we tested the commercially available DAGLB antibodies. Unfortunately, none of them stained midbrain DANs. However, RNAscope® in situ hybridization demonstrated the co-localization of *Daglb* and *Dagla* mRNA with the dopamine synthase *tyrosine hydroxylase* (*Th*) in mouse nigral DANs (Fig. 2d, Supplementary Fig. 9). Besides *Th*-positive DANs, both *Daglb* and *Dagla* mRNA signals were also detected in the *Th*-negative cells in SNc and *substantia nigra pars reticulata* (SNr) (Fig. 2d, Supplementary Fig. 9). Therefore, while *DAGLA* mRNA is highly expressed by most neurons in the brain[22–24], *DAGLB* mRNA is the main 2-AG synthase expressed by nigral DANs, suggesting a nigral DAN-specific mechanism of *DAGLB*-deficiency in the pathogenesis of Parkinsonism.

**Daglb knockdown in nigral DANs reduces 2-AG levels in the SN.** To examine the functional significance of DAGLB, we decided to selectively knockdown (KD) *Daglb* or *Dagla* in nigral DANs using an adeno-associated virus (AAV)-based CRISPR/SaCas9 genome editing system (AAV-CMV-DIO-SaCas9-U6-sgRNA)[31]. The control (Ctrl) saCas9 empty (referred to as the AAV-Ctrl) and saCas9 with the *Daglb* or *Dagla*-guided sgRNA (referred to as the AAV-*Daglb* KD and AAV-*Dagla* KD) are expressed in a Cre-dependent manner (Fig. 3a). Co-transfection of AAV-Cre and AAV-*Daglb* KD vectors led to substantial reduction of DAGLB but not DAGLA protein levels in primary cultured mouse cortical and hippocampal neurons (Fig. 3b–d). Similarly, co-transfection of AAV-Cre and AAV-*Dagla* KD vectors specifically suppressed the expression of DAGLA protein (Fig. 3b-d). As a control, the co-transfection of AAV-Cre and AAV-Ctrl did not affect the expression of DAGLB or DAGLA protein (Fig. 3b-d). Therefore, we developed *Daglb*- and *Dagla*-specific gene targeting vectors to selectively disrupt the expression of *Daglb* and *Dagla*, respectively, in a Cre-dependent manner.

To measure 2-AG release in the SN in vivo, we stereotaxically injected the AAVs carrying a genetically encoded eCB sensor named eCB2.0[32,33] in the dorsal striatum (Fig. 3e, f). A custom-

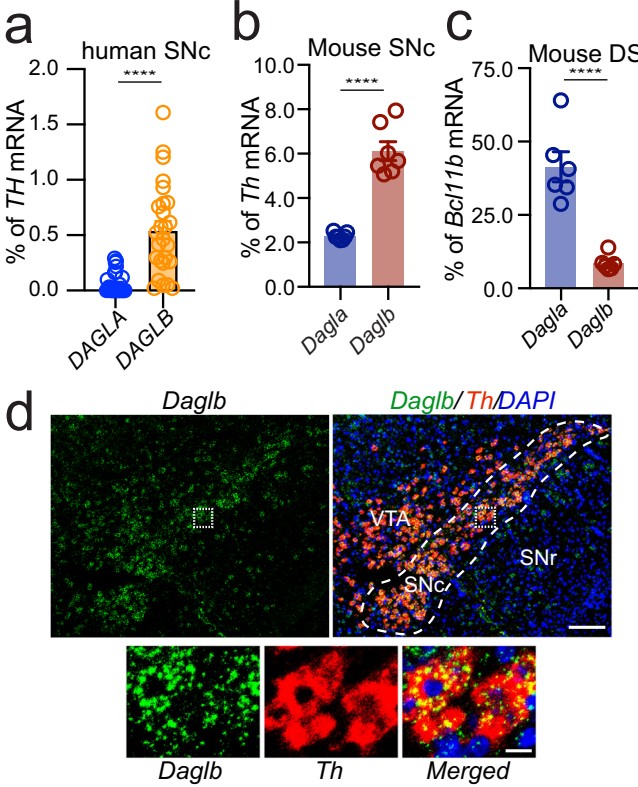

**Fig. 2 DAGLB is the main 2-AG synthase expressed by the nigral DANs.**
**a** Quantification of *DAGLA* and *DAGLB* mRNA expression by RNA-sequencing of LCM-isolated human nigral DAN samples ($n = 26$)[29].
**b** Quantification of *Dagla* and *Daglb* mRNA expression by RNA-sequencing of LCM-isolated nigral DANs from adult mouse brains ($n = 7$). **c** *Dagla* and *Daglb* mRNA expression in the mouse dorsal striatum (DS, $n = 6$). The expression of *Dagla* and *Daglb* mRNAs was normalized by the expression of *Th* mRNAs in the nigral DANs and *Bcl11b* mRNAs in the SPNs. Data were presented as mean ± SEM. Unpaired *t* test, two-tailed, ****$p < 0.0001$.
**d** RNAscope® in situ hybridization of *Daglb* and *Th* in mouse midbrain sections. Sections were counterstained with DAPI. Dashed line outlines the SNc region. Bottom panels highlight the boxed areas in the up panels. SNc: substantia nigra pars compacta. SNr: substantia nigra pars reticulata. VTA: ventral tegmental area. Scale bars: 100 μm (left) and 20 μm (right). More than five independent experiments were performed.

built fiber photometry system[34] was employed to capture the eCB2.0 signals in the SNr area (Fig. 3e), where the axons of dSPNs and dendrites of DANs form synaptic connections[11,14]. The infusion of control (Ctrl)-, *Daglb* KD-, or *Dagla* KD-AAVs in the SNc of DAT^IRES^*Cre* mice leads to selective expression of either saCas9 empty (referred to as DAN-Ctrl) or saCas9 with the *Daglb*- or *Dagla*-sgRNA (referred to as DAN-*Daglb* KD or *Dagla* KD) in the DANs (Fig. 3e). We used hemagglutinin (HA) staining to visualize the distribution of HA-tagged saCas9 in different brain regions (Supplementary Fig. 10a, b). Overall, around 75% of DANs in the SNc were infected with AAV-Ctrl or AAV-*Daglb* KD (Supplementary Fig. 10c). We did not observe any HA staining in other brain regions besides the midbrain. We also did not find any loss of nigral DANs 12 months after AAV injection (Supplementary Fig. 10d). Additionally, we could not detect any apparent alterations of dopamine and other bioamine contents in the dorsal striatum, neuroinflammation, axon swelling, or mitochondrial deformation in the DAN-*Daglb* KD mice 4-6 months after stereotaxic surgery (Supplementary Fig. 10e–h).

The eCB2.0 sensor was constructed based on CB1 receptors, of which the third intracellular loop is replaced with circularly

permuted green fluorescent protein (cpGFP) and the binding of 2-AG and AEA enhances the emission intensity of cpGFP[33] (Fig. 3f). Like the native CB1 receptor, eCB2.0 sensors were transported to the dSPN axon terminals in the SN (Fig. 3g). The AAVs carrying red fluorescent protein tdTomato (tdT) were co-injected with AAV-eCB2.0 as a reference for adjusting any motion artifacts during the imaging process[34]. Two distinct emission peaks corresponding to the eCB2.0 and tdT signals were detected in the SNr immediately before and 120 min after the administration of JZL184 (16 mg/kg), a selective monoacylglycerol lipase inhibitor[35] that blocks the degradation of 2-AG (Fig. 3h). The JZL184-induced enhancement of eCB2.0 signals was substantially diminished in the DAN-*Daglb* KD mice compared to the controls in both time- and dose-dependent manner [time: two-way ANOVA, $F(7,42) = 60.3$, $p < 0.0001$; dose: two-way ANOVA, $F(3,18) = 66.7$, $p < 0.0001$] (Fig. 3i, j). Furthermore, liquid chromatography-tandem mass spectrometry (LC-MS/MS) also revealed a marked reduction of 2-AG levels in the SNc of DAN-*Daglb* KD mice (unpaired *t* test, $p = 0.01$) (Fig. 3k). By contrast, genetic deletion of *Dagla* in the SN DANs did not affect the JZL184-induced enhancement of eCB2.0 signals in the SNr (Supplementary Fig. 11). Together, these results demonstrate that DAGLB is the dominant 2-AG synthase that catalyzes the 2-AG production in nigral DANs.

**The SN 2-AG levels correlate with motor performance during locomotor skill learning.** Our recent study demonstrates that genetic ablation of nigral DANs in mouse models only modestly reduced the walking speed, but completely abolished the improvement of motor performance in the rotarod locomotor skill learning test[8,36], revealing the involvement of nigral DAN activity in locomotor skill learning. We thereby examined the 2-AG signals in the SNr by simultaneously conducting fiber photometry live recording and the rotarod training, with 10 trials per daily session for six consecutive days (Fig. 4a). Each trial started with a low constant rotating speed at 4 rpm for 30 s before linear acceleration from 4 to 40 rpm in 5 min[37]. The 2-AG signals gradually increased along with the progression of 10 trials during each training session (Fig. 4a, b, Supplementary Fig. 12). In addition, more robust daily enhancement of 2-AG levels was recorded on the first four days of training compared to the last two days of training [one-way ANOVA, $F(5,42) = 12.2$, $p < 0.0001$] (Fig. 4c). The first four days of training are generally regarded as the acquisition phase of motor learning, while the last two days of training are regarded as the maintenance or retention phase[38,39]. As expected, rotarod performance was also greatly improved during the acquisition phase, but not in the retention phase (Fig. 4d). Indeed, further correlational analyses in which the mean eCB2.0 signals in each of the 10 trials on each training session and the corresponding rotarod performance (latency to fall) were used to calculate the correlation coefficients. The analysis reveals stronger positive correlations between the 2-AG signal enhancement and rotarod performance in the acquisition phase compared to the retention phase [one-way ANOVA, $F(5,42) = 16.9$, $p < 0.0001$] (Fig. 4e). These results suggest that the SN 2-AG signaling is particularly engaged in the improvement of motor performance during the acquisition phase of locomotor skill learning.

***Daglb* knockdown in the nigral DANs compromises the dynamic 2-AG release during the early phase of locomotor skill learning and impairs the overall motor performance.** To further examine the contribution of *Daglb*-mediated 2-AG biosynthesis in the nigral DANs during locomotor skill learning, we selectively knocked down the expression of *Daglb* in nigral DANs of 3-

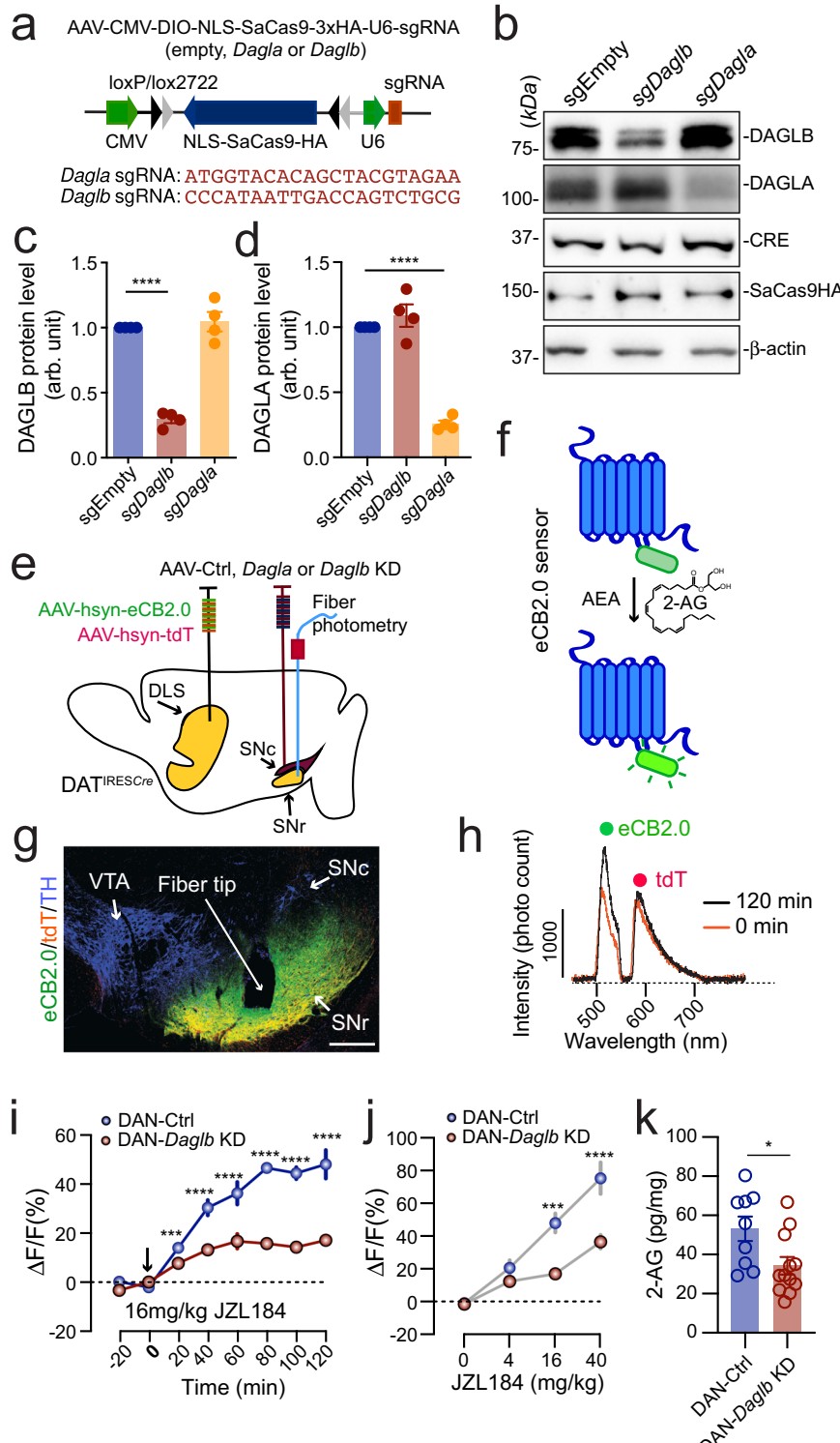

month-old DAT[IRESCre] mice with AAV vectors and compared the SN 2-AG levels in DAN-Ctrl and DAN-*Daglb* KD mice during rotarod tests (Fig. 5a). We found that the increase of eCB2.0 signals was substantially diminished in the SN of DAN-*Daglb* KD mice compared to the DAN-Ctrl mice, especially in the later trials on days 1 through 5 across the training sessions as compared to the DAN-Ctrl ones (Fig. 5b). To examine the behavioral consequence of diminished 2-AG signaling in DAN-*Daglb* KD mice, we compared the rotarod performance between DAN-Ctrl and DAN-*Daglb* KD mice. Although the DAN-*Daglb*

KD mice seemed to perform worse than the DAN-Ctrl mice, the difference was not statistically significant due to the small sample size (five per genotype) [two-way ANOVA, genotype: $F(1, 8) = 1.85$, $p = 0.2100$] (Supplementary Fig. 13a). We then generated a separate large cohort of DAN-Ctrl and DAN-*Daglb* KD mice ($n = 16$ per genotype) specifically for motor behavioral tests. DAN-*Daglb* KD mice showed markedly impairments in the rotarod locomotor skill learning test compared to the control DAN-Ctrl mice (Fig. 5c). We further compared the rotarod performance between the first trials of each training session, as

**Fig. 3 DAGLB mediates the main 2-AG synthesis in the nigral DANs. a** Diagram of AAV-mediated CRISPR/saCas9 gene targeting vector and the sequence of *Dagla* and *Daglb* sgRNAs. **b** Western blot of DAGLB in cultured cortical and hippocampal neurons transfected with control, *Dagla*, or *Daglb* KD AAV vectors in combination with Cre-expressing AAV vectors. Actin was used as a loading control. Four independent experiments were performed. **c, d** Bar graph quantifies DAGLB (**c**) and DAGLA (**d**) protein levels in four independent experiments. Data were presented as mean ± SEM. One-way ANOVA, multiple comparisons, adjusted [****]$p < 0.0001$. **e** Schematic illustrates fiber photometry recording of 2-AG signals in the SN of DAN-control and DAN-*Daglb* KD mice. DLS: dorsolateral striatum. **f** Cartoon of eCB sensor eCB2.0. **g** Co-staining of eCB2.0, tdT and TH in the midbrain sections of DAN-*Daglb* KD mice. Scale bar: 100 μm. More than five independent experiments were performed. **h** Sample photon counts of eCB2.0 (wavelength: 500–540 nm) and tdT (wavelength: 575–650 nm) emission immediately before and 120 min after JZL184 (16 mg/kg) administration. **i** Time course of eCB2.0 signals in the SN of DAN-control [$n = 4$, 2 Male(M)/2 Female(F)] and DAN-*Daglb* KD ($n = 4$, 2M/2F) mice before and after JZL184 (16 mg/kg) treatment. Data were presented as mean ± SEM. Sidak's multiple comparisons, adjusted [***]$p = 0.0001$. [****]$p < 0.0001$. **j** Dose response of eCB2.0 signals 120 min after JZL administration at 0 (vehicle only), 4, 16, and 40 mg/kg. $n = 4$ mice per group. Data were presented as mean ± SEM. Sidak's multiple comparisons, adjusted [***]$p = 0.0003$. [****]$p < 0.0001$. **k** LC-MS/MS quantification of 2-AG in the SNc of DAN-control ($n = 9$) and DAN-*Daglb* KD ($n = 12$) mice. Data were presented as mean ± SEM. Unpaired $t$ test, two-tailed, [*]$p = 0.02$.

well as the last trials (Fig. 5d). During the first trial, the rotarod performance was significantly improved across the training sessions in DAN-Ctrl mice (one-way ANOVA, $F = 4.344$, $p = 0.0014$) and performance on days 3, 5, and 6 was significantly better than performance on day 1 [Multiple comparisons, adjusted $p$ value (day 1 vs. days 2–6) = 0.3418, 0.0004, 0.0604, 0.0043, and 0.0103, respectively]. By contrast, there were no significant changes in the rotarod performance in the first trials in DAN-*Daglb* KD mice (one-way ANOVA, $F = 0.5854$, $p = 0.7111$) and performance on days 2–6 was comparable to performance on day 1 [Multiple comparisons, adjusted $p$-value (day 1 vs. day 2–6) = 0.4439, 0.5520, 0.6926, 0.7463, and 0.9633, respectively]. Moreover, the rotarod performance in the first trials on days 3 through 6 was significantly worse in the DAN-*Daglb* KD mice compared to the control mice (Multiple $t$ test, $p = 0.0009$, 0.02, 0.0007, and 0.0006, respectively) (Fig. 5d), implying deficiency in retention or consolidation of learned locomotor skills across training sessions. Contrary to the first trials in each training session, the rotarod performance in the last trials was largely comparable across the training sessions in DAN-Ctrl mice (one-way ANOVA, $F = 2.095$, $P = 0.0732$). However, the performance on day 6 was significantly better than performance on day 1 [multiple comparisons, adjusted $P$ -value (day 1 vs. day 2–6) = 0.1391, 0.3302, 0.0553, 0.0868, and 0.0215, respectively]. On the one hand, the DAN-*Daglb* KD mice, like the DAN-Ctrl mice, showed no significant changes in the overall rotarod performance in the last trials of each training session (one-way ANOVA, $F = 2.177$, $P = 0.0636$), although the performance on day 6 was better than performance on day 1 [Multiple comparisons, adjusted $P$-value (day 1 v. days 2–6) = 0.9253, 0.4052, 0.1832, 0.4400, and 0.0138, respectively]. On the other hand, the performance of DAN-*Daglb* KD mice in the last trials on days 2, 4, and 5 was significantly worse than the performance of DAN-Ctrl mice in the corresponding training sessions (Multiple $t$ test, $p = 0.0050$, 0.0369, and 0.0191, respectively) (Fig. 5d). Except for the impaired locomotor skill learning, there were no apparent alterations of spontaneous locomotor activity nor gait properties in the DAN-*Daglb* KD mice compared to the controls (Supplementary Fig. 13b, c). Together, these results demonstrate that DAGLB-mediated 2-AG biosynthesis in nigral DANs is actively engaged in regulating the functional role of nigral DANs in locomotor skill learning, especially during across-session learning.

**Genetic deletion of CB1 receptor in dSPNs impairs rotarod locomotor skill learning.** Since the dSPNs are also required for rotarod locomotor skill learning[40] and provide main inhibitory inputs to the postsynaptic nigral DANs[10], we next examined whether CB1 receptors in the axon terminals of dSPNs mediate the retrograde 2-AG signaling during locomotor skill learning. To selectively delete the CB1 receptor-encoding *Cnr1* gene in the

dSPNs, we crossbred a line of *Cnr1*-floxed (*Cnr1*[fl/fl]) mice[41] with dopamine receptor D1-Cre (*Drd1*-Cre) mice. Accordingly, the expression of CB1 receptors was completely abolished in the axon terminals of dSPNs in *Drd1*-Cre/ *Cnr1*[fl/fl] conditional knockout (CB1 cKO) mice (Fig. 6a). Like the DAN-*Daglb* KD mice, the CB1 cKO mice also displayed substantial impairment in rotarod locomotor skill learning compared to the control mice (Fig. 6b). We also conducted further analyses on the rotarod performance in the first and last trials in both CB1 control and cKO mice (Fig. 6c). The rotarod performance in the first trials was significantly improved across the training sessions in control *Cnr1*[fl/fl] mice (one-way ANOVA, $F = 7.711$, $P < 0.0001$). The performance on days 2 through 6 was significantly better than performance on day 1 [multiple comparisons, adjusted $P$ value (day 1 vs. days 2–6) = 0.0229, 0.0010, 0.0002, 0.0001, and <0.0001, respectively]. Significant improvement in the first trials was also observed in the CB1 cKO mice (one-way ANOVA, $F = 5.792$, $P = 0.0002$). Additionally, the performance on days 3 through 6 was better than performance on day 1 [Multiple comparisons, adjusted $P$-value (day 1 vs. days 2–6) = 0.4999, 0.0035, 0.0088, 0.0036, and <0.0001, respectively]. However, the performance of CB1 cKO mice in the first trials on days 2, 4, 5, and 6 was significantly worse than the control mice (Multiple $t$ test, $p = 0.0474$, 0.0375, 0.0458, and 0.0204, respectively) (Fig. 6C). Similar to the first trials, while the performance in the last trials of each training session was significantly improved in both CB1 control and cKO mice (one-way ANOVA, *Cnr1*[fl/fl] mice: $F = 4.403$, $P = 0.0026$; CB1 cKO mice: $F = 3.125$, $P = 0.0130$), the performance of CB1 cKO mice in the last trials on days 3 through 6 was significantly worse than the control mice (Multiple $t$ test, $p = 0.0232$, 0.0435, 0.0387, and 0.0031, respectively) (Fig. 6C). Together, these results demonstrate that the disruption of CB1 receptor-mediated eCB signaling in dSPNs affects rotarod locomotor skill learning, particularly during cross-session locomotor skill learning, supporting the notion that the nigral DAN-derived 2-AG regulates locomotor skill retention and consolidation at least partly through modulating the presynaptic inputs from dSPNs.

**Daglb germline knockout mice do not develop any overt motor behavioral and neuropathological abnormalities.** Like *DAGLB*, the loss-of-function mutations in *PARKIN*, *DJ-1*, and *PINK1* also contribute to the etiopathogenesis of PD[5]. However, the *Parkin*, *Dj-1*, and *Pink1* germline KO mice failed to develop any profound motor impairments or severe nigral DAN loss[42]. Similarly, we did not observe any apparent alterations of locomotor activity in *Daglb* germline KO mice at 4, 8, 12, and 20 months of age (Supplementary Fig. 14a–c). Since the rotarod locomotor learning test provides a more sensitive behavioral paradigm to detect the dysfunction of nigral DANs[8], we examined the rotarod performance of *Daglb* germline KO mice at 4 and 20 months of age. The

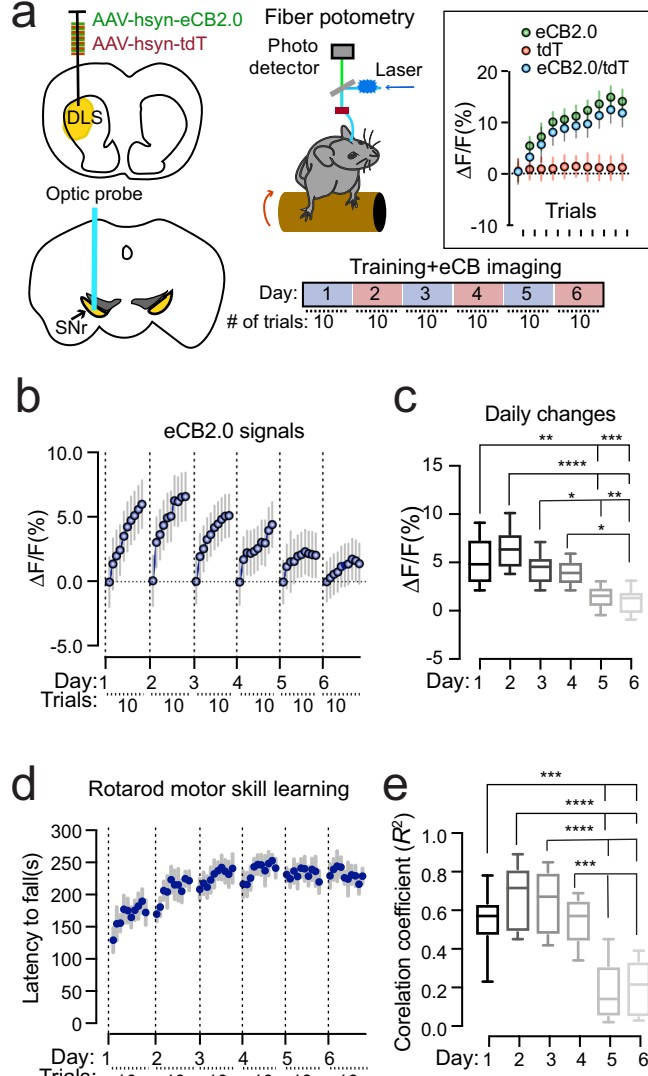

**Fig. 4 DAGLB-mediated 2-AG biosynthesis in the nigral DANs is involved in locomotor skill learning. a** Schematic diagram of eCB2.0 fiber photometry recoding while mice undergo rotarod locomotor skill training. Inset shows the average eCB2.0 (green), tdT (red), and normalized eCB2.0 (blue) signals during each trial from one representative mouse on day 2 of the 6-day training paradigm ($n = 1$). Data were presented as mean ± SD. **b** Normalized eCB2.0 signal intensity during the locomotor learning. $n = 8$ (4M/4F). Data were presented as mean ± SD. **c** Box and whiskers plot (the minimum, the 25th percentile, the median, the 75th percentile, and the maximum) of maximal daily increase of eCB2.0 signal intensity. $n = 8$ (4M/4F). Tukey's multiple comparisons adjusted, $^*p = 0.02$, $^{**}p = 0.001$, $^{***}p = 0.0003$, $^{****}p < 0.0001$. **d** Performance of rotarod motor skill training. $n = 8$ (4M/4F). Data were presented as mean ± SEM. **e** Box and whiskers plot (the minimum, the 25th percentile, the median, the 75th percentile, and the maximum) of correlation coefficient of eCB2.0 signal intensity and rotarod performance. $n = 8$ (4M/4F). Tukey's multiple comparisons adjusted, $^{***}p = 0.0002$ to 0.0006, $^{****}p < 0.0001$.

4-month-old *Daglb* KO mice displayed modest but statistically insignificant improvement of locomotor learning [two-way ANOVA, genotype: $F(1, 19) = 2.9$, $p = 0.103$] (Supplementary Fig. 14d), while the 20-month-old *Daglb* KO mice showed similar performance compared to the littermate controls [two-way ANOVA, genotype: $F(1, 19) = 0.02$, $p = 0.899$] (Supplementary Fig. 14e). Additionally, no apparent loss of TH-positive nigral

DANs was found in the 20-month-old *Daglb* KO mice [Unpaired *t* test, $p = 0.9989$] (Supplementary Fig. 4f). Therefore, the CRISPR/SaCas9-mediated acute knockdown of *Daglb* in the nigral DANs of adult mice provide a more sensitive experimental paradigm to evaluate the contribution of DAGLB-dependent 2-AG signaling in nigral DANs during locomotor learning.

**2-AG signaling potentiates nigral DAN activity and somatodendritic dopamine release.** Since 2-AG from nigral DANs acts on the presynaptic CB1 receptors to suppress the release of inhibitory neurotransmitter GABA from dSPN axon terminals[20], we suspected that the JZL184-induced 2-AG upregulation in the SN (Fig. 3i, j) may disinhibit the presynaptic inhibitory inputs from dSPNs and lead to enhanced DAN activity and somatodendritic dopamine release. To test this hypothesis, we first treated the mice with JZL184, and then used fiber photometry with genetically encoded calcium indicator GCaMP6f[43] and dopamine indicator DA2m[44] to monitor the DAN activity and somatodendritic dopamine release. To examine the DAN calcium transients, we crossbred DAT[IRESCre], Ai95 (RCL-GCaMP6f) and Ai9 (RCL-tdT) mice to selectively express GCaMP6f and tdT in the midbrain DANs of DAT[IRESCre]/GCaMP6f/tdT triple transgenic mice, and then stereotaxically injected AAV-Ctrl or AAV-*Daglb* KD vectors in the SNc of triple transgenic mice to evaluate the role of DAGLB in regulating DAN activity (Fig. 7a, b). The intraperitoneal injection of JZL184 (20 mg/kg) led to significant increase of DAN activity as indicated with the elevated GCaMP6f signal intensities in the SNc of both DAN-Ctrl [two-way ANOVA, treatment: $F(1, 4) = 36.89$, $p = 0.0037$] and DAN-*Daglb* KD [two-way ANOVA, treatment: $F(1, 4) = 36.88$, $p = 0.0037$] triple transgenic mice compared to the vehicle treatment (Fig. 7c). However, the JZL184 treatment induced significantly less robust enhancement of neuronal activity [two-way ANOVA, genotype: $F(1, 4) = 14.58$, $p = 0.0188$] in the SNc of DAN-*Daglb* KD compared to the DAN-Ctrl triple transgenic mice (Fig. 7c).

To monitor dopamine release in the SN of DAN-Ctrl and DAN-*Daglb* KD mice, we stereotaxically injected AAV-DAm2 and AAV-tdT vectors in the dorsal striatum, and AAV-Ctrl or AAV-*Daglb* KD vectors in the SNc of DAT[IRESCre] mice (Fig. 7d, e). The same JZL184 treatment also significantly enhanced dopamine release as indicated by the increased DA2m fluorescent signal intensities in the SN of both DAN-Ctrl [two-way ANOVA, treatment: $F(1, 8) = 23.13$, $p = 0.0013$] and DAN-*Daglb* KD [two-way ANOVA, treatment: $F(1, 6) = 18.7$, $p = 0.0049$] mice compared to the vehicle treatment. While the JZL184-induced dopamine release was not statistically significant between DAN-*Daglb* KD and DAN-Ctrl mice during the entire 60 min period [two-way ANOVA, genotype: $F(1, 7) = 4.612$, $p = 0.0689$], multiple comparisons showed significantly lower dopamine levels in the SN of DAN-*Daglb* KD mice 10, 20, and 50 min after the drug treatment (Fig. 7f). Taken collectively, these data suggest a dynamic interplay between the dopamine and 2-AG signaling in the nigral DANs, in which the DAGLB-mediated 2-AG biosynthesis in nigral DANs promotes the DAN activity and somatodendritic dopamine release.

**Inhibition of 2-AG degradation rescues the motor impairments of *Daglb*-deficient mice.** Since the JZL184 treatment enhanced nigral DAN activity and dopamine release in both the control and DAN-*Daglb* KD mice (Fig. 7c, f), and the activity of nigral DANs is essential for locomotor skill learning[8,36], we then treated a cohort of 4-month-old DAN-Ctrl and DAN-*Daglb* KD mice with JZL184 or vehicle 1 h before each day's rotarod locomotor training session. The JZL184 treatment (20 mg/kg) significantly improved the locomotor skill learning in both DAN-

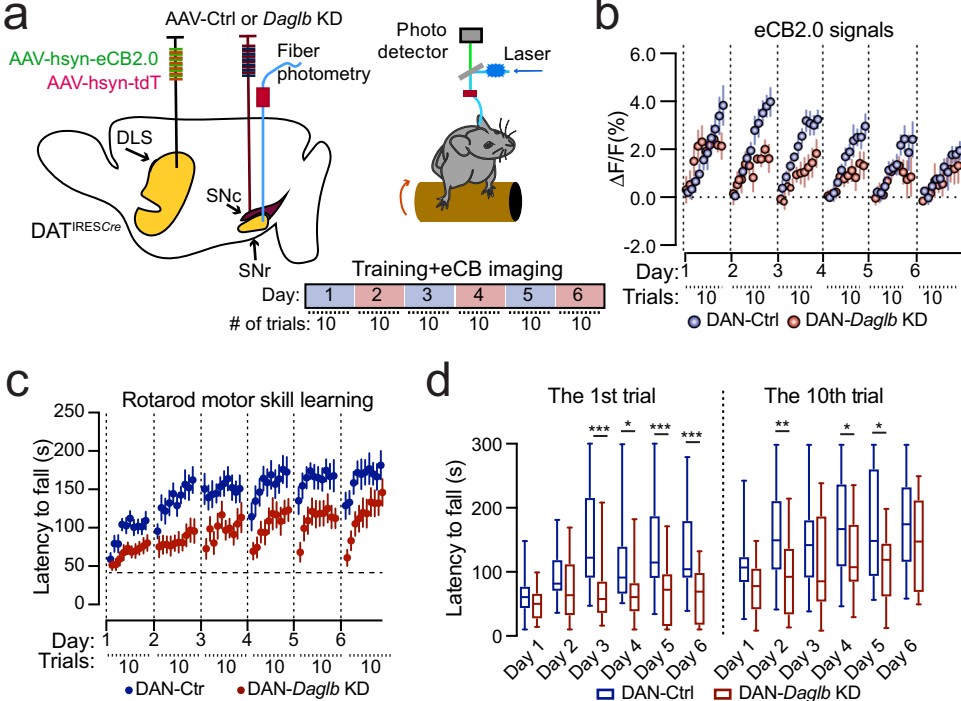

**Fig. 5 *Daglb*-deficiency in nigral DANs compromises the early dynamic release of 2-AG and impairs rotarod locomotor skill learning. a** Schematic diagram of eCB2.0 fiber photometry recoding while DAN-Ctrl or DAN-*Daglb* KD mice undergo rotarod locomotor skill training. **b** Normalized eCB2.0 signal intensity over the course of locomotor learning of DAN-control and DAN-*Daglb* KD ($n = 5$ M per genotype) mice. Data were presented as mean ± SEM. Two-way ANOVA, genotype: $F(1, 8) = 13.68$, $p = 0.006$. **c** Rotarod locomotor skill training of a different cohort of DAN-control ($n = 16$, 8M/8F) and DAN-*Daglb* KD ($n = 16$, 8M/8F) mice. Data were presented as mean ± SEM. Two-way ANOVA, genotype: $F(1, 30) = 9.3$, $p = 0.0047$. **d** Box and whiskers plot (the minimum, the 25th percentile, the median, the 75th percentile, and the maximum) of rotarod performance in the first and 10th trial on each day with the same cohorts of DAN-control ($n = 16$, 8 M/8F) and DAN-*Daglb* KD ($n = 16$, 8M/8F) mice. Multiple unpaired $t$ test, two-tailed, $^*p < 0.05$, $^{**}p < 0.005$, $^{***}p < 0.001$.

*Daglb* KD mice [two-way ANOVA, treatment: $F(1, 21) = 59.9$, $p < 0.0001$] and DAN-Ctrl mice [two-way ANOVA, treatment: $F(1, 15) = 18.3$, $p < 0.0001$] compared to the vehicle-treated mice (Fig. 8a). Moreover, the administration of JZL184 completely rescued the locomotor learning deficits of DAN-*Daglb* KD mice, significantly improving their performance relative to the vehicle-treated DAN-Ctrl mice [two-way ANOVA, genotype: $F(1, 17) = 6.645$, $p = 0.0196$] (Fig. 8a). Therefore, the blockage of 2-AG degradation by JZL184 is sufficient to restore the local 2-AG levels required for the rotarod locomotor skill learning in DAN-*Daglb* KD mice.

## Discussion

In the present work, we identified four different Parkinsonism-causal loss-of-function mutations in *DAGLB* and showed that DAGLB is the dominant 2-AG synthase in nigral DANs. Supporting the physiological importance of *DAGLB*-mediated 2-AG biosynthesis in nigral DAN-dependent motor functions, we found that genetic KD of *Daglb* in the mouse nigral DANs led to reduced SN 2-AG levels and impaired rotarod locomotor skill learning. Additionally, pharmacological inhibition of 2-AG degradation increased SN 2-AG levels, promoted DAN activity and dopamine release, and rescued the motor deficits. Therefore, we reveal a DAN-specific pathophysiological mechanism of *DAGLB* dysfunction in the pathogenesis of Parkinsonism and provide the rationale and preclinical evidence for the potential beneficial effects of 2-AG augmentation in alleviating Parkinsonism[45].

High levels of eCBs were detected in the cerebrospinal fluid of untreated patients with PD[46]. Increased eCB levels in the globus

pallidus are associated with reduced movement in a PD animal model[47]. However, previous studies in patients with PD and related animal models mostly focus on the alterations of eCB signaling after severe nigral DAN loss or lengthy levodopa administration[26]. The results are thereby more likely to reflect the compensatory responses to the disease. It was unclear, however, whether the changes of eCB signaling contribute to the etiopathogenesis of the disease. Our human genetics study provides genetic evidence to demonstrate that like dopamine deficiency, the impairment of 2-AG signaling also contributes to the pathogenesis of Parkinsonism. *DAGLB* is not linked to sporadic PD cases of European descent; however, one caveat to using AMP-PD data is that it mainly includes sporadic late-onset PD cases that are not enriched for monogenic or early onset cases. This may explain why it is always hard to use AMP-PD to replicate rare familial mutations. *DAGLB* is a gene duplication of *DAGLA*[24]. Although DAGLA is the dominant 2-AG synthase in most neurons and accounts for 80% of 2-AG production in the CNS[22,23], our gene expression and functional assays demonstrate that DAGLB mediates the major 2-AG biosynthesis in nigral DANs. The predominant presence of *DAGLB* in nigral DANs may explain why the loss-of-function mutations in *DAGLB* lead to DAN dysfunction and Parkinsonism. On the other hand, the elevation of 2-AG levels in the other brain regions as observed in the patients with PD[46] likely represents a compensatory response to the loss of *DAGLB*-mediated 2-AG production in the nigral DANs due to PD-related dopaminergic neurodegeneration.

Although some modest loss of nigral DANs and other neuropathological and motor behavioral abnormalities were observed in the aged (>2-year-old) *Parkin* germline KO mice[48], *Daglb*

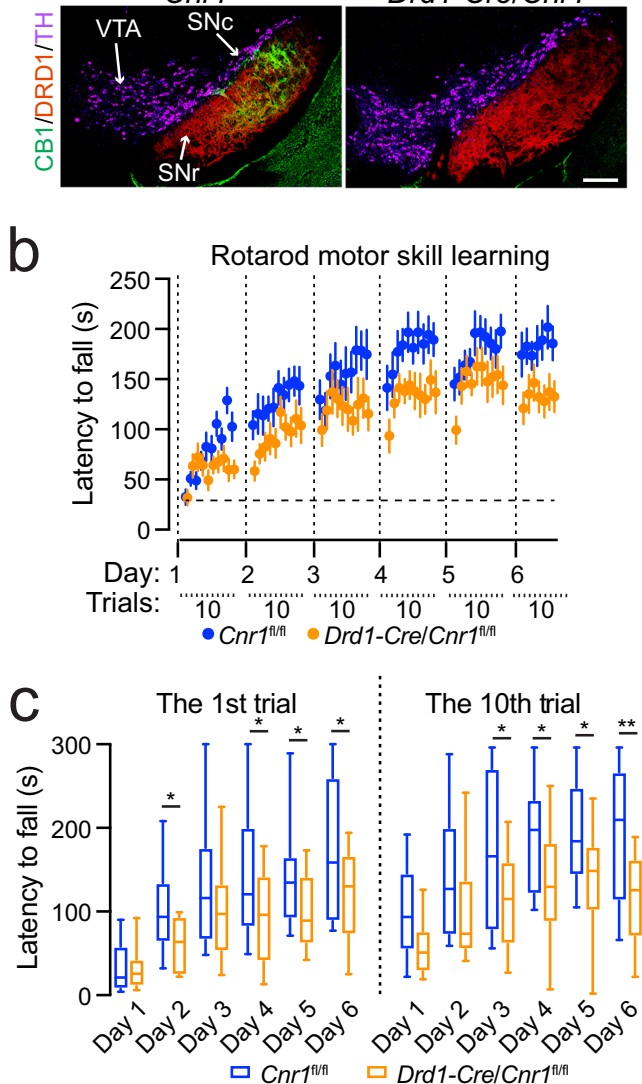

**Fig. 6 Genetic deletion of CB1 receptor in dSPNs impairs rotarod locomotor skill learning. a** Co-staining of CB1 (green), DRD1 (red), and TH (purple) in the midbrain sections of homozygous floxed CB1 (*Cnr1*^fl/fl) and Drd1-Cre/*Cnr1*^fl/fl mice. Scale bar: 100 μm. More than five independent experiments were performed. **b** Rotarod locomotor skill training of *Cnr1*^fl/fl ($n = 14$, 7 M/7F) and Drd1-Cre/*Cnr1*^fl/fl ($n = 12$, 6M/6F) mice. Data were presented as mean ± SEM. Two-way ANOVA, genotype: $F(1, 24) = 6.301$, $p = 0.0192$. **c** Box and whiskers plot (the minimum, the 25th percentile, the median, the 75th percentile, and the maximum) of rotarod performance in the first and 10th trial on each day with the same cohorts of *Cnr1*^fl/fl ($n = 14$, 7M/7F) and Drd1-Cre/*Cnr1*^fl/fl ($n = 12$, 6M/6F) mice. Multiple unpaired *t* test, two-tailed, \*$p < 0.05$, \*\*$p < 0.005$.

germline KO mice failed to develop any apparent neuropathological and motor behavioral phenotypes at 20 months of age. Longer lifespan and other genetic and physiological characteristics may render human neurons more susceptible to the disease-related mutations[49]. To overcome the difficulty in studying the impact of PD-related recessive mutations on nigral DANs with germline KO mice and avoid any potential compensatory interference during development, we applied CRISPR/saCas9-mediate knockdown of *Daglb* selectively in the nigral DANs of adult DAT^IRESCre mice. We also subjected the DAN-*Daglb* KD mice to the nigral DAN-sensitive rotarod locomotor skill learning test to

examine any nigral DAN dysfunction. Finally, we employed fiber photometry live recording technique to monitor the 2-AG release in behaving mice during rotarod locomotor performance. Together, we offer an alternative experimental scheme to study the pathophysiological mechanism of Parkinsonism-related genetic mutations in mouse models and reveal a nigral DAN-specific pathogenic mechanism of *Daglb*-deficiency in Parkinsonism. Since the expression of saCas9 is Cre-dependent and the viral vectors were stereotaxically injected in the SNc of DAT^IRESCre mice, the saCas9 was mainly expressed by the nigral DANs. The AAV9 serotype possesses very limited capacity in retrograde axonal transport[50]. However, even if the AAV9 vector could effectively get into the axons, none of those SNc-projecting neurons express DAT. For example, the *locus coeruleus* nor-epinephrinergic neurons express TH but not DAT. DAT is only expressed by the DANs. Our observations are in line with the initial characterization of DAT^IRESCre mice, in which the Cre-dependent gene expression is only observed in the midbrain DANs and to a lesser extent in the olfactory glomeruli[51]. Those olfactory dopaminergic interneurons do not extend their axons outside of olfactory bulb. Therefore, with the combination of DAT^IRESCre mice and local infusion of Cre-dependent viral vectors in the SNc, the *Daglb* KD is constrained to the midbrain DANs.

Since the overall efficiency of CRISPR/saCas9-mediated *Daglb* knockdown is about 70-80% in the current study, the remaining 25% of nigral DANs still possess intact DAGLB activity and may contribute to the increase of 2-AG levels during rotarod loco-motor training and after JZL184 administration. Additional sources of 2-AG could also come from the production of DAGLA in nigral DANs, as well as both DAGLA and DAGLB from non-dopaminergic neurons in the SNc and SNr. Our findings are consistent with a recent study[32], in which the presence of JZL184 strongly suppressed the degradation of 2-AG and allowed 2-AG to diffuse for longer distance and act on the neighboring CB1 receptors for longer duration. A line of *Daglb* conditional KO mice that selectively delete *Daglb* in all the adult nigral DANs would be useful to reveal potentially more severe behavioral and neurochemical phenotypes. In addition, DAGLA, although a minor 2-AG synthase in nigral DANs, may also contribute to the residual 2-AG production in *Daglb*-deficient DANs. Genetic deletion of both *Dagla* and *Daglb* in all nigral DANs may provide the means to critically evaluate the pathophysiological role of 2-AG in nigral DAN-dependent motor behaviors. Noticeably, the intensities of 2-AG signals were not strictly corelated with the motor performance especially during trials 6-10 in each rotarod session. While the 2-AG levels continued to build up over sequential trials, the improvement of motor performance apparently was not linear. The complex relationship between 2-AG transmission and motor performance echoes findings in changes of neuronal ensemble activity in the dorsal striatum during rotarod locomotor learning[39]. This interesting observation likely indicates that the peak rotarod performance is determined at the organism level and by multiple neural circuits and factors.

DAGLA protein is enriched in dendritic spines[22,23]; however, the subcellular localization of DAGLB protein remains unclear due to a lack of specific antibodies for tissue staining. Many factors could contribute to the difficulty in immunostaining. The relatively low expression level of DAGLB protein in nigral DANs could be a factor. Additionally, compared to the western blot, in which the proteins were denatured and unfolded, the DAGLB proteins may exist as the native folded conformation in tissue sections, which could potentially mask the antibody binding sites and reduce the accessibility of antibody. Considering that the CB1-positive axon fibers form close contact with the dendrites and cell bodies of ventral nigral DANs[13] and 2-AG acts within a

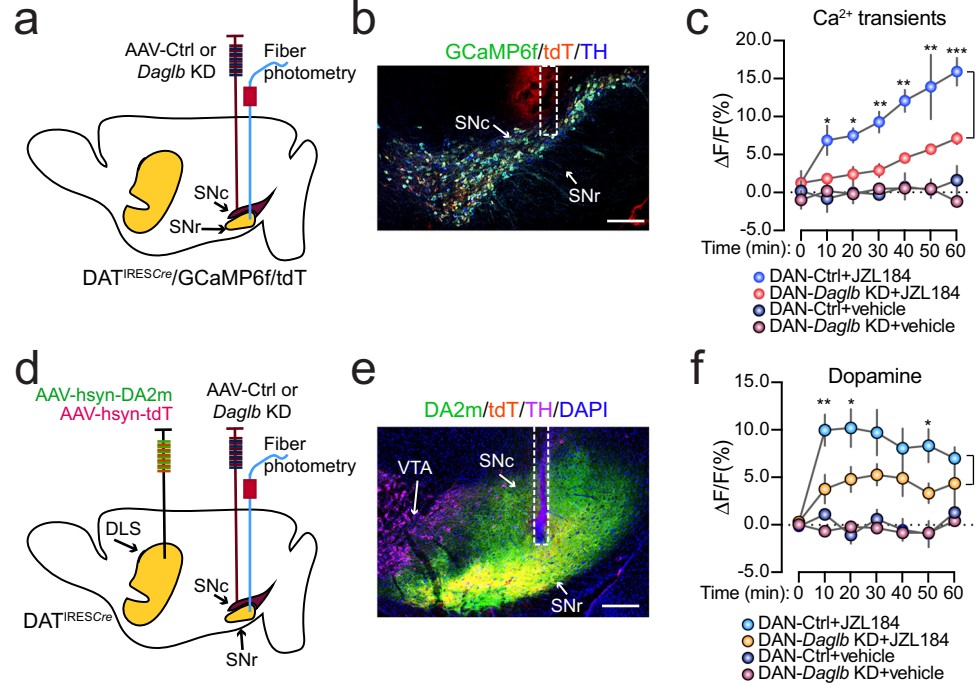

**Fig. 7 JZL184 treatment promotes DAN activity and dopamine release. a** Schematic illustrates fiber photometry recording of GCaMP6f and tdT signals in the SNc of DAN-control and DAN-*Daglb* KD Dat[IRESCre]/GCaMP6f/tdT trigenic mice. **b** Representative images of GCaMP6f (green), tDT (red), and TH (blue) staining. Scale bar: 200 μm. Six independent experiments were performed. **c** Alterations of calcium transients in the SNc of DAN-control and DAN-*Daglb* KD Dat[IRESCre]/GCaMP6f/tdT trigenic mice ($n = 3M$ per genotype) treated with vehicle or JZL184 (20 mg/kg). Data were presented as mean ± SEM. Multiple unpaired *t* test, two-tailed, $^*p < 0.05$, $^{**}p < 0.01$, $^{***}p < 0.001$. **d** Schematic illustrates fiber photometry recording of DA2m and tdT signals in the SN of DAN-control and DAN-*Daglb* KD mice. **e** Representative images of DA2m (green), tdT (red) and TH (magenta) staining. Scale bar: 200 μm. Nine independent experiments were performed. **f** Changes of dopamine release in the SN of DAN-control ($n = 5 M$) and DAN-*Daglb* KD ($n = 4 M$) mice treated with vehicle, JZL184 (20 mg/kg). Data were presented as mean ± SEM. Multiple unpaired *t* test, two-tailed, $^*p < 0.05$, $^{**}p < 0.01$.

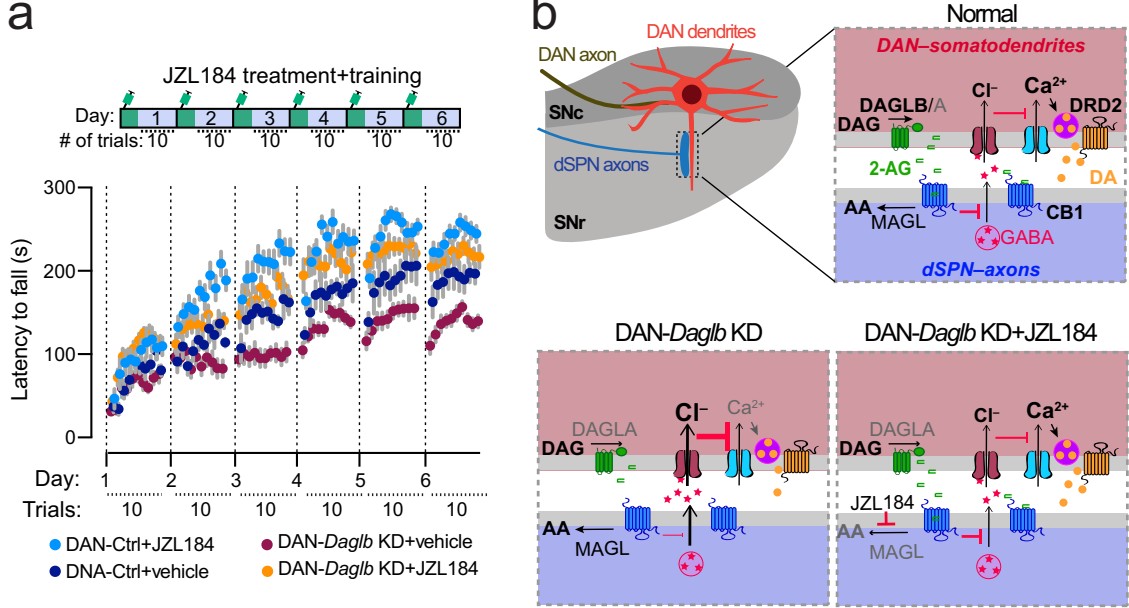

**Fig. 8 JZL184 treatment rescues the locomotor skill learning impairments of DAN-*Daglb* KD mice. a** Rotarod locomotor skill learning of DAN-control ($n_{vehicle} = 7$, 4M/3F; $n_{JZL184} = 10$, 5M/5F) and DAN-*Daglb* KD ($n_{vehicle} = 11$, 6M/5F; $n_{JZL184} = 12$, 6M/6F) mice treated with vehicle or JZL184 (20 mg/kg). Data were presented as mean ± SEM. **b** A working model of nigral DAN-derived 2-AG signaling in regulating DAN neuron activity and dopamine release in SN region, where 2-AG modulates DAN calcium influx and somatodendritic dopamine release through retrograde inhibition of presynaptic GABA release from dSPNs via CB1-mediated intracellular signaling pathway. Genetic deletion of *Daglb* in DANs compromises the 2-AG-mediated feedback regulation, while pharmacological inhibition of 2-AG degradation by JZL184 restores the signaling. Our study suggests that this nigral DAN-derived 2-AG signaling in the SN is involved in regulating the across-session locomotor skill learning. DAG diacylglycerol, DA dopamine, AA arachidonic acid.

short range (~10 μm) from the release sites[32,52], it is reasonable to assume that DAGLB is distributed in the soma and dendrites of DANs for local 2-AG production and release (a working model in Fig. 8b). The rotarod locomotor training paradigm appears to promote a more pronounced somatodendritic release of 2-AG from the nigral DANs more pronouncedly during the early phase of locomotor learning. The elevated 2-AG likely acts on the presynaptic CB1 receptors to suppress the release of the inhibitory neurotransmitter GABA from the dSPN axon terminals[20], resulting in the enhanced DAN firing and dopamine release that is required for the locomotor performance and learning process. By contrast, the loss-of-function of *Daglb* in nigral DANs dampens the dynamic enhancement of 2-AG release especially during the acquisition of locomotor skills and compromises locomotor performance. Accordingly, a previous study also demonstrated that the administration of eCB agonist Δ9-tetrahydrocannabinol increases the DAN firing rate, dopamine synthesis, and dopamine release in dopaminergic axon terminals in striatum[53], while the CB1 receptor agonist WIN55,212-2 induces a dose-dependent increases in firing rate and burst firing in nigral DANs[54]. Considering that genetic deletion of postsynaptic CB1 receptors in dSPNs leads to similar rotarod locomotor learning impairments as the presynaptic inhibition of 2-AG production in nigral DANs, the *DAGLB*-mediated 2-AG production in nigral DANs may enhance the nigral DAN activity and somatodendritic dopamine release and facilitate the locomotor learning through attenuating the inhibitory inputs from dSPNs. Since 2-AG works locally near the production and release sites, we only examined the interplay between 2-AG and dopamine signaling in the nigral regions. Future experiments will be needed to investigate whether the change of 2-AG release in the soma and dendrites of nigral DANs affects the dopamine release in DAN axon terminals at dorsal striatum.

Although the rotarod locomotor training paradigm promotes on-demand 2-AG production by DAGLB, how the DAGLB activity is regulated remains to be determined. Striosome dSPN axons are intermingled with the dendrites of aldehyde dehydrogenase 1a1 (ALDH1A1)-positive DANs perpendicularly protruding in the SNr and form so-called striosome-dendron bouquet structures[11,12,14]. The ALDH1A1-positive nigral DANs display distinct rebound activity in response to the inhibitory inputs from dSPNs[11,12], which then trigger large dendritic Ca$^{2+}$ transients likely through T-type Ca$^{2+}$ channels[12,55]. Future studies will be needed to selectively knockout *Daglb* in the ALDH1A1-positive DANs and evaluate whether the intracellular Ca$^{2+}$ elevation in dendrites is required to induce on-site 2-AG production and release, which in turn retrogradely suppresses the GABAergic inhibition from dSPNs and further accelerates the rebound activity of nigral DANs. Besides the transsynaptic action, 2-AG can also function cell-autonomously within the DANs through promoting both the pace-maker activity and evoked burst firing[56], suggesting that *DAGLB*-deficiency in nigral DANs could also lead to reduced DAN activity and associated motor impairments. Therefore, the nigral DANs can produce and release both dopamine and 2-AG, while 2-AG may further boost the dopamine release and neural activity in response to increasing demand. While our study suggests that the JZL184-induced nigral dopamine release depends on the presence of nigral DANs, the efficacy of JZL184 treatment could be weakened in PD cases with severe loss of nigral DANs. Nonetheless, the administration of JZL184 would be still beneficial to compensate for the loss of 2-AG signaling in patients who carry the *DAGLB* mutations, as well as to enhance dopamine release in patients who have enough nigral DANs remaining.

Although the present study focused on the neuronal function of 2-AG, this eCB has also been implicated in inflammation[57].

Myeloid synthesis of 2-AG appears to promote vascular inflammation and atherogenesis[58]. *Daglb* inactivation in mouse peritoneal macrophages attenuates lipopolysaccharide-induced release of proinflammatory cytokine tumor necrosis factor-α[59]. Since the inhibition of DAGLB activity works against inflammatory responses[59], we reason that the *DAGLB*-deficiency is less likely to directly induce the harmful neuroinflammation implicated in the pathogenesis of PD. Nonetheless, future studies will be needed to further elucidate the role of *DAGLB* in microglia or other non-neuronal cells in PD.

In conclusion, our study supports an involvement of *DAGLB*-mediated 2-AG biosynthesis in regulating the normal physiological function of nigral DANs, which may help to explain how *DAGLB*-deficiency contributes to PD-related motor symptoms. To boost the production of 2-AG may thereby serve as a potential mechanistic-based therapeutic intervention to enhance nigral dopamine release and neuronal activity in patients who preserve enough nigral DANs. Indeed, an exploratory clinical trial of eCB-like cannabidiol seems to improve the mobility and mental states of patients with PD[45].

## Methods

**Study participants**. Participants were recruited at Xiangya Hospital, Central South University between October 2006 and December 2021 and other hospitals of Parkinson's Disease and Movement Disorders Multicenter Database and Collaborative Network in China (PD-MDCNC, http://pd-mdcnc.com) established by our group. These participants include 171 probands with ARPD [86 men (50.29%), mean age at onset: 55.55 ± 11.39], 1,571 cases of sporadic EOPD [865 men (55.06%); mean age at onset, 44.00 ± 5.57], and 2,152 matched healthy control participants (mean age, 61.78 ± 11.70)[27]. All participants were subjected to the standard clinical neurological examination. PD was diagnosed according to the UK Parkinson's disease Society Brain Bank clinical diagnostic criteria[60] or Movement Disorders Society (MDS) clinical diagnostic criteria for Parkinson's disease[61] by at least two neurologists. The healthy participants did not have any nervous system or psychiatric diseases. Human blood samples and fibroblasts were obtained after participants provided written informed consent. All investigations were conducted according to the Declaration of Helsinki, and the study was approved by the Institutional Review Boards of the Ethics Committee of Xiangya Hospital, Central South University. All participants consented to participation in the study (see also ref. [27]). In addition to consenting to participate in the study the six patients with DAGLB mutations described agree to the publication of this study.

**PET Study**. According to a previously reported method[62], positron emission tomography/computed tomography (PET/CT) was performed on the Family 2 II-4 using [11]C-**2**β-carbomethoxy-**3**β-(**4**-fluorophenyl) tropane ([11]C-CFT) tracer. Before the PET/CT imaging, the patient discontinued the drug intake for 2 days to avoid the potential effect of anti-PD drugs. Brain PET imaging was performed at 1 h after intravenous injection of [11]C-CFT. The regions of interest in each hemisphere were identified and drawn on the caudate nucleus, putamen, and cerebellum.

**SNP genotyping and homozygosity mapping**. DNA samples of Family 1 (AR-003), Family 2 (AR-005), and Family 4 underwent genome-wide SNP array genotyping. Genome-wide genotyping was performed with the Illumina Human Omni ZhongHua-8 Bead Chip arrays. Homozygosity mapping was performed with PLINK (https://zzz.bwh.harvard.edu/plink/) for the identification of regions of homozygosity in affected individuals, and the minimum length for homozygous runs was set to 2 Mb.

**Whole-exome sequencing**. Exome data were obtained from the 171 cases with ARPD, 1,571 cases of sporadic EOPD, and 1,652 matched control participants. As previously reported[27], whole-exome DNA was capture using the SureSelect Human All Exon Kit V5 or V6 (Agilent) and high-throughput sequencing was conducted using the Illumina X10 with a coverage more than 100 X. Burrow-Wheeler Aligner was implemented to align Paired-end sequence reads onto the reference human genome (UCSC hg19). The Picard tool (http://broadinstitute.github.io/picard/) was used to remove duplicate reads, generate the converse format, and index the sequencing data. Base quality-score recalibration, local realignments around possible insertions/deletions (indels), variant calling, and filtering were performed with the Genome Analysis Toolkit (GATK)[63]. ANNOVAR[64,65] was used to annotate single nucleotide variants and insertions/deletions with RefSeq (UCSC hg19), such as gene regions, amino acid alterations, functional effects, and allele frequencies in East Asian population from GnomAD database and ExAC database. Mutations of previously reported PD causative genes were excluded. The minor allele frequency of the variants was limited to 0.01 for the above population database. Only

predicted damaging missense and loss-of-function variants (nonsense variants, frameshift indels, and splicing-site variants) were included.

**Sanger sequencing**. Potential mutations were confirmed by Sanger sequencing and were shown to segregate with the phenotype. Potential compound heterozygous variants were ruled out based on the location in the same chromosome and Sanger sequencing of different family members. Mutation analysis of *DAGLB* in another 500 matched control cohort was done by direct sequencing (GenBank, NM_139179.4 and NP_631918.3). Genomic DNAs from individuals were amplified by PCR with oligonucleotide primers complementary to flanking intronic sequences. Samples were run and analyzed on an ABI PRISM 3130 genetic analyzer (Applied Biosystems).

**Detection and validation of CNVs in *DAGLB***. The detection of copy-number variant (CNV) in *DAGLB* from WES data in our ARPD and sporadic EOPD cohorts was performed with the eXome-Hidden Markov Model (XHMM) software, which uses principal component analysis normalization and a hidden Markov model to detect and genotype CNVs from normalized read-depth data in targeted sequencing experiments. To further analyze the detailed structure variants of *DAGLB* in our patients identified by the WES CNV analysis, we used the Oxford Nanopore platform to sequence the same individuals (Family 3, AR-075). As previously reported[66,67], large insert-size libraries were created according to the manufacturer recommended protocols (Oxford Nanopore). Libraries were sequenced on R9.4.1 flow cells using PromethION. NGMLR and Sniffles were used to analyze structural variations. All reads were aligned to the human reference genome (hg19) using NGMLR (ngmlr 0.2.7), and structural variation calls were detected by Sniffles. Candidate structural variations were subjected to manual examination and further validation. The Sanger sequencing of the PCR product of the breakpoints was performed using standard protocols.

**Search of AMP-PD datasets**. We used coding regions from whole-genome sequencing (WGS) data from the Accelerating Medicines Partnership - Parkinson's disease initiative (AMP-PD, https://amp-pd.org/), which includes data from multiple cohorts of unrelated European descent[68]. All data was in GRCh38 format. Excluding any individuals recruited through genetic enrichment arms and non-PD diagnoses, our analysis included 4,007 people (1,451 PD cases and 2,556 controls). Variants were classified into 2 classes (missense and LOF), as annotated using ANNOVAR[64] (2020-06-08). PLINK[69] was used to extract missense and LOF (splicing, stop, frameshift) variants at a MAF of 0.05, and to run genotypic disease-variant association testing, using the–model flag. After recoding the data into an additive model, summary statistics were extracted using R (version 4.1.0). Finally, RVtests[70] (v2.1) was used to run the CMC, CMC Wald, and SKAT-O burden tests for each of the defined variant classes at MAF of 0.05. The use of AMP-PD datasets follows the ethical oversight given for AMP-PD.

**Human skin fibroblast culture**. Human dermal fibroblasts were derived from skin biopsies from affected individuals and age- and sex-matched non-neurological controls, through standard techniques[71]. Fibroblasts were cultured in Dulbecco's modified Eagle's medium (ThermoFisher) supplemented with 10% fetal bovine serum, penicillin, and streptomycin (Gibco). Cells were grown in 5% $CO_2$ at 37 °C in a humidified incubator. To further investigate the protein stability of DAGLB in human dermal fibroblasts, cells were treated with proteasome inhibitor MG132 (Sigma, M7449) at 10 μM for 24 h before the cells were collected. DMSO was used as a control vehicle.

**Quantitative RT–PCR (qRT–PCR)**. Total RNA was extracted with RNeasy kit (QIAGEN), and first-strand cDNA synthesis was performed with a SuperScript III First-Strand Synthesis system (Invitrogen). Real-time Taqman PCR was performed on an ABI 7900HT with TaqMan Gene Expression Assays (Applied Biosystems, Life Technologies, Carlsbad, CA) for human *DAGLB* exon 9-10 (Hs00373700_m1). Results were normalized to GAPDH. Experiments were performed with triplicate experimental samples and controls, and fold increases were calculated using the comparative threshold cycle method.

**Western blotting**. Cells were lysed in radioimmunoprecipitation assay (RIPA) buffer containing protease and phosphatase inhibitor cocktails and sonicated for 1 min with the Bioruptor sonication device (Diagenode). Cell lysates were centrifuged at $13,000 \times g$ for 10 min at 4 °C, and the supernatant was collected for protein quantification (Pierce BCA Protein Assay Kit). Each sample contained 20 μg of proteins and was mixed with Bolt LDS Sample Buffer and Sample Reducing Agent (ThermoFisher) and heated at 70 °C for 10 min. The prepared protein extracts were size fractioned by 4–12% NuPAGE Bis-Tris gel electrophoresis (Invitrogen) using MES running buffer (Invitrogen). After transfer to the nitrocellulose membranes using Transfer Cell (Bio-Rad), the membranes were blocked with Odyssey Blocking Buffer (LI-COR) and probed overnight with the appropriate dilutions of the primary antibodies. The antibodies used for western blot analysis included Rabbit monoclonal anti-DAGLB (Cell Signaling, 12574S, clone D4P7C, 1:500), Rabbit polyclonal anti-DAGLA (Frontier Institute co. ltd,

DGLa-Rb-Af380, 1:250), Rabbit monoclonal anti-HA-Tag (Cell Signaling, 3724S, clone C29F4, 1:500), Rabbit monoclonal anti-Cre Recombinase (Cell Signaling, clone D7L7L, 12830S, 1:500), Mouse monoclonal anti-GAPDH (Sigma-Aldrich, G8795, clone GAPDH-71.7, 1:5000) and Mouse monoclonal anti-β-actin (Sigma-Aldrich, A2228, clone AC-74, 1:5000). Incubation with the IRDye-labeled secondary antibodies (LI-COR, 1:10000) was performed for 1 h at room temperature. The protein bands of interest were visualized with Odyssey CLx Infrared Imaging Studio. The band intensity was quantified using ImageJ.

**Mouse work**. All mouse studies were in accordance with the guidelines approved by the Animal Care and Use Committees (IACUC) of the Intramural Research Program of National Institute on Aging (NIA), National Institutes of Health (NIH). The wild-type C57BL/6J (#000664), DAT[IRESCre] (#006660), Ai95 (RCL-GCaMP6f) (#028865), and Ai9 (RCL-tdT) (#007909) mice were purchased from the Jackson laboratory. *Cnr1*[loxP/loxP] mice[41] were provided by Dr. Josephine M. Egan of NIA. *Daglb* germline KO mice[22] were provided by Dr. Ku-Lung Hsu of University of Virginia. Mice were housed at temperature of ~18–20 °C with 40–60% humidity in a twelve-hour-light/twelve-hour-dark cycle and were fed water and regular diet ad libitum. All the behavioral tasks were performed during the light cycles. The genotype, gender, and age of mice were indicated in the figure legends.

**RNA in situ hybridization with RNAscope**. RNAscope (Advanced Cell Diagnostics, ACD) was performed according to the manufacturer's instructions on fresh frozen tissue sections. The sample preparation and pretreatment were conducted according to the instructions of RNAscope Multiplex Fluorescent Reagent Kit v2 user manual. RNAscope probes for *Daglb* (Cat No. 497801-C1), *Dagla* (Cat No. 478821-C3), and *Th* (Cat No. 317621-C2) were purchased from ACD and used according to the company's online protocols. Fluorescent images were acquired using a laser scanning confocal microscope LSM 780 (Zeiss).

**RNA-sequencing**. For the RNA sequencing of nigrostriatal DANs, we used LCM to isolate GFP-positive DANs in the SNc region of *Pitx3*[+/IRES2-tTA] (JAX#021962)/ pTRE-H2BGFP (JAX#005104) double transgenic mice as described previously[72]. The animals were anesthetized with $CO_2$ followed by decapitation at one year old. The brains were rapidly dissected and frozen in dry ice. The frozen brains were sectioned at 30 μm thickness by a cryostat onto a PAN membrane frame slide (Applied Biosystems, Foster City, CA) and stored at −80 °C until LCM performance. By an ArturusXT microdissection system with fluorescent illumination (Applied Biosystems), the GFP-positive cells in the SNc region were selected and then captured onto LCM Macro Caps (Applied Biosystems) at the following working parameters: spot size, 7–25 μm; power, 50–70 mW; duration, 20–40μs. The total RNA was extracted and purified with the PicoPure Isolation kit (Applied Biosystems) and genomic DNA was cleaned-up by RNase free DNase (Qiagen) after the protocols provided by the manufacturers. The RNA was quantified using a NanoDrop spectrophotometer (ThermoFisher) and the RNA integrity was measured using the Bioanalyzer RNA 6000 pico assay (Agilent). The cDNA libraries were generated from the purified RNA using TruSeq Stranded Total RNA LT library preparation kit (Illumina) according to the manufacturer's instructions. The libraries were then qualified using the Bioanalyzer DNA 1000 assay (Agilent) and sequenced with Illumina HiSeq 2000. The standard Illumina pipeline was used to generate Fastq files. The Ensembl annotated transcript abundance was quantified using Salmon in a non-alignment-based mode, and gene-level counts were estimated using Tximport package (Bioconductor). The counts for the resulting genes were then normalized using a variance-stabilizing transformation. For the RNA sequencing of striatal tissues, the dorsal striatum of 3-month-old C57BL/6 J mice were dissected and subjected to RNA extraction, sequencing and data analysis as described recently[30]. The accession number of the striatal tissue RNA-seq is PRJNA612478.

The accession number of SNc RNA-seq data is PRJNA775656.

**AAV-*Daglb* KD gene targeting vector construction, validation, and packaging**
*Vector construction*. All plasmids were constructed using standard recombinant DNA cloning techniques. The PX601-AAV-CMV::NLS-SaCas9-NLS-3xHA-bGHpA;U6::BsaI-sgRNA plasmid was a gift from Feng Zhang (Addgene plasmid # 61591)[31]. The *Daglb* sgRNA oligos were designed with Benchling (https://benchling.com) and subcloned into the PX601-AAV-CMV::NLS-SaCas9-NLS-3xHA-bGHpA;U6::BsaI-sgRNA vector. To construct a single AAV vector harboring both Cre-dependent SaCas9 transgene and constitutively expressed sgRNA expression cassette, the Magneto2.0-sNRPpA element of pAAV-CMV-DIO-Magneto2.0-sNRPpA expression vector, a gift from Ali Guler (Addgene plasmid # 74307)[73], was replaced by the SaCas9-NLS-3xHA-bGHpA;U6::BsaI-sgRNA DNA fragment.

*Vector validation*. Neuro-2a (N2a) cell lines were maintained in Dulbecco's modified Eagle's medium (DMEM) supplemented with 10% FBS (HyClone), 2 mM GlutaMAX (Life Technologies), 100U/ml penicillin, and 100 mg streptomycin at 37 °C with 5% $CO_2$ incubation. Cells were co-transfected with pAAV-Cre-GFP and pAAV-CMV-DIO-SaCas9-NLS-3xHA-bGHpA;U6::BsaI-sgRNA plasmids or

pAAV-EF1a-DIO-mCherry plasmids (Addgene plasmid #50462) at the ratio of 1:3 using X-tremeGENE HP DNA Transfection Reagent (Roche) following the manufacturer's recommended protocol. Cells were harvested for PCR-based identification of mutations caused by genome editing using the Guide-it Mutation Detection Kit (Cat. No. 631443), and immunoblotting was performed to analyze the DAGLB protein levels.

*AAV packaging*. The packaging was carried out by a commercial source (Vigene Biosciences Inc.) and the resulting AAVs had titers of $1.0 \times 10^{13}$ to $2.0 \times 10^{14}$ genome copies per ml.

*Stereotaxic injection*. The stereotaxic survival surgery was performed as previously described[8]. 500 nL of AAVs with titers $8.0 \times 10^{13}$ genome copies per ml were loaded into 2 μL Neuros Syringes (Hamilton) and were injected into brain areas at chosen coordinates. The coordinates based on Bregma coordinates, and the coordinates for SNc are AP -3.1 mm, ML: ± 1.5 mm, DV −3.9 mm.

**Primary neuronal culture and viral infection**. Mouse primary neuronal cultures were prepared from the cortices of embryonic day 16.5 embryos. Briefly, cortices were dissected in cold Hank's balanced salt solution, and incubated with 0.025% trypsin for 20 min at 37 °C. The digested tissue was triturated into single cells using glass Pasteur pipettes and filtering through 70 μm nylon cell strainer. The cells were seeded and plated in Biocoat Poly-D-Lysine Cellware plate and maintained in neurobasal medium supplemented with 2% B27 and 2 mM GlutaMax at 37 °C in 5% $CO_2$ humidified incubator. Cells at 4 days in vitro (DIV) were infected with AAV DJ-Cre-GFP and AAV DJ-CMV-DIO-SaCas9-NLS-3xHA-bGHpA;U6::BsaI-sgRNA at the ratio of 1:3, and the cells were harvested 7 days after infection for western blot analyses.

**Behavioral tests**

*Rotarod locomotor skill learning test*. Mice were placed onto a rotating rod with auto acceleration from 4 to 40 rpm in 5 min (Panlab). The duration that each mouse was able to stay on the rotating rod in each trial was recorded as the latency to fall. The standard motor learning task was performed as ten trials per day for six consecutive days, as described previously[8].

*Open-field test*. The ambulatory, rearing, and fine movements of mice were measured with the Photobeam Activity System (San Diego Instruments). PAS software was used to trace and quantify mouse movement in the unit as the number of beam breaks per 30 min, as previously described[74].

*Gait analysis*. The Free Walk Scan System (CleverSys Inc) was used for gait analysis, as described before[8]. Briefly, mice were allowed to move freely in a 40 cm × 40 cm × 30 cm (length × width × height) chamber. A high-speed camera below a clear bottom plate was used to capture mouse movement for 5 min in the red light. Videos were analyzed using FreewalkScanTM2.0 software (CleverSys Inc) for various characteristic parameters of gait, including stride length and stance/swing time of each paw.

**Histology, immunohistochemistry, and light microscopy**. Mice were anesthetized with ketamine and then transcardially perfused with 4% PFA/PBS solution. Brains were isolated, post-fixed in 4% PFA overnight, and then submerged in 30% sucrose for 72 h at 4 °C for later sectioning. Series of 40 μm sections were collected using a cryostat (Leica Biosystems). Sections were blocked in 10% normal donkey serum, 1% bovine serum albumin, 0.3% Triton X-100, and PBS solution for overnight at 4 °C. The sections were then incubated with the primary antibodies over one to two nights at 4 °C. The antibodies used for immunostaining included rat monoclonal anti-DRD1 (Sigma-Aldrich, D2944, clone 1-1-F11 s.E6, 1:500), mouse monoclonal anti-CB1 (Synaptic systems, 258011, clone 289c1, 1:500), rabbit polyclonal anti-TH (Pel-Freez Biologicals, P40101, 1:2500), mouse monoclonal anti-TH (ImmunoStar, 22941, 1:1000), chicken polyclonal anti-TH (Aves Labs, TYH, 1:500), chicken polyclonal anti-GFP (Aves Labs, GFP-1020, 1:1000), rabbit polyclonal anti-RFP (Rockland, 600-401-379, 1:1000), rabbit monoclonal anti-HA-Tag (Cell Signaling, 3724 S, clone 29F4, 1:100), guinea pig polyclonal anti-NeuN (Synaptic systems, 266 004, 1:1000), rabbit polyclonal anti-Iba1 (Wako, 019-19741, 1:1000), and mouse monoclonal anti-TOM20 (Santa Cruz, sc-17764, clone F-10, 1:200). Sections were then washed three times in PBS before being incubated in the secondary antibody solutions with Alexa Fluor 488, 546, or 633-conjugated IgG (H + L) Cross-Adsorbed secondary antibodies (1:500, Invitrogen) at 4 °C for overnight. Following three washes in PBS, sections were mounted onto subbed slides, and coverslipped with mounting media (ProLong® Gold Antifade Mountant, Life technology). The stained sections were imaged using a laser scanning confocal microscope (LSM 780, Zeiss) with Zen software. The paired images in the figures were collected at the same gain and offset settings.

**Stereology**. According to the mouse brain in stereotaxic coordinates, a series of coronal sections across the midbrain (40 μm per section, every fourth section from Bregma −2.54 to −4.24 mm, ten sections per case) were processed for TH

immunohistochemistry and finally visualized using a laser scanning confocal microscope (LSM 780, Zeiss). The images were captured as a single optic layer under ×20 objective lens. TH-positive neurons in SNc were assessed using the fractionator function of Stereo Investigator 10 (MBF Bioscience) as described previously[72]. Five mice were used per group. Counters were blinded to the genotypes of the samples.

**2-AG measurement with liquid chromatography-tandem mass spectrometry**. Endocannabinoids were extracted from the SNc of 3 to 4-month-old mice and quantified by LC-MS/MS as previously described[75]. In brief, the fresh brain tissues were sliced at 500 μm thickness and frozen immediately in liquid nitrogen. The samples were taken by punch technique then kept on dry ice or at −80 °C. Tissue samples from individual mice were homogenized in 80-300 μl of Tris buffer (pH 8.0) and the protein concentrations were determined by Bradford assay. Ice-cold methanol/Tris buffer (50 mM, pH 8.0) solution was added to each homogenate (1:1, vol/vol). 200 ng [$^2H_5$] of arachidonoyl glycerol ([$^2H_5$]2-AG) was used as internal standard. The homogenates were extracted three times with $CHCl_3$:methanol (2:1, vol/vol), dried under nitrogen and reconstituted with methanol after precipitating proteins with ice-cold acetone. The dried samples were reconstituted in 50 μl of ice-cold methanol, and 2 μl of which were analyzed with liquid chromatography in line mass spectrometry. The LC-MS/MS analyses were conducted on an Agilent 6410 triple quadrupole mass spectrometer (Agilent Technologies) coupled to an Agilent 1200 LC system. Analytes were separated using a Zorbax SB-C18 rapid-resolution HT column. Gradient elution mobile phases consisted of 0.1% formic acid in water (phase A) and 0.1% formic acid in methanol (phase B). Gradient elution (250 mL/min) was initiated and held at 10% B for 0.5 min, followed by a linear increase to 85% B at 1 min and maintained until 12.5 min, then increased linearly to 100% B at 13 min and maintained until 14.5 min. The mass spectrometer was set for electrospray ionization operated in positive ion mode. The source parameters were as follows: capillary voltage, 4000 V; gas temperature, 350 °C drying gas, 10 L/min; nitrogen was used as the nebulizing gas. Collision-induced dissociation was performed using nitrogen. Level of each compound was analyzed by multiple reactions monitoring. The molecular ion and fragment for each compound were measured as follows: m/z 348.3/91.1 for [$^2H_5$]2-AG and m/z 379.3/91.1 for 2-AG. Analytes were quantified by using Mass-Hunter Workstation LC/QQQ Acquisition and MassHunter Workstation Quantitative Analysis software (Agilent Technologies). Levels of 1-arachidonoyl-glycerol (1-AG) and 2-AG in the samples were measured against standard curves. Since 2-AG is chemically unstable in aqueous solutions and some of them were transformed to 1-AG during the sample preparation and detection, we used the sum of 2-AG and 1-AG concentration as the total 2-AG concentration in tissues.

**In vivo fiber photometry**. A custom-built dual-color fiber photometry system[34] was used for in vivo measurement of eCB2.0, DA2m, GCaMP6f, and tdTomato fluorescent signals. For imaging the eCB2.0 or DA2m signals, 600 nl AAV9-hsyn-eCB2.0 ($2.3 \times 10^{12}$ GC/ml, Vigene Biosciences) or AAV9-hsyn-DA2m ($4.67 \times 10^{11}$ GC/ml, Vigene Biosciences) AAVs were mixed with 200 nl AAV9-hsyn-tdTomato ($1.38 \times 10^{12}$ GC/ml, Vigene Biosciences) AAVs and stereotaxically injected in the dorsal striatum (coordinates: AP + 0.5 mm, ML + 2.4 mm, DV -2.5 mm; AP + 1.5 mm, ML + 1.8 mm, DV-3.0 mm) of 3 to 4-month-old wild-type C57BL/6J (JAX#000644) or DAT$^{IREScre}$ (JAX#006660) mice. Four weeks after viral injection, an optical probe (200 μm core and 0.22 NA) was implanted with the tips sitting in the SNr areas (coordinates: AP-3.16 mm, ML + 1.4 mm, DV −4.4 mm) for imaging the eCB2.0 or DA2m signals. The animals were allowed to recover for at least one week after the fiber implantation surgery before the fiber photometry measurement. The fluorescence signals were acquired using 49 ms integration time and were triggered by 20 Hz transistor-transistor logic pulses from an output pulse generator. The eCB2.0, DA2m, and GCaMP6f fluorescence signals were calculated by total photo counts between 500 nm and 540 nm. The tdTomato fluorescence signals were calculated by total photon counts between 575 nm and 650 nm. The measured emission spectra of eCB2.0, DA2m, GCaMP6f, and tdTomato signals were fitted using a linear unmixing algorithm (https://www.niehs.nih.gov/research/atniehs/labs/ln/pi/iv/tools/index.cfm). The coefficients of eCB2.0, DA2m, GCaMP6f and tdTomato signals generated by the unmixing algorithm were used to represent the fluorescence intensities of eCB2.0, DA2m, GCaMP6f and tdTomato, respectively. To correct for movement-induced artifacts, the ratios of eCB2.0, DA2m or GCaMP6f signal intensities against their corresponding tdTomato signal intensities were used to represent the final normalized signal intensities.

For the JZL184 experiments, fiber photometry recordings were conducted in free-moving animals for 20 min before drug administration to measure the baseline fluorescent intensities and then for 60 min or 120 min after drug treatment. The average baseline signals were calculated as $F_B$. The instant signals at different time point after drug treatment were calculated as $F_I$. The alterations of signal intensities at different time points were calculated as $\Delta F/F = (F_I - F_B)/F_B$.

The rotarod locomotor skill learning and fiber photometry recoding experiments were performed as reported previously[37]. Briefly, in each trial the mice were put on a rotatable rod (EZRod, Omnitech Electronics) starting at 4 rpm constant speed for 30 s and then steadily accelerated from 4 to 40 rpm in 5 min, while the fiber photometry recording was performed at the same time. 10 trials and recordings were carried out each day for six consecutive days. $F_B$, the baseline signal intensity of

the first trial of each day, is the average signal intensities at 4 rpm for the first 30 s. $F_I$ represents the average signal intensity during each trial. The alterations of signal intensities at different trials were calculated as $\Delta F/F = (F_I - F_B)/F_B$. The correlation of eCB2.0 signal and rotarod performance was calculated on a trial-by-trial basis. For each mouse on a given day, the mean eCB signals in each of the 10 trials and the corresponding rotarod performance (latency to fall) were used to calculate the correlation coefficient ($R^2$). The average $R^2$ of eight mice on each day was plotted on the graph.

**Dissection of striatal and midbrain tissue from mouse brain slices**. Mice were anesthetized by intraperitoneal injection of pentobarbital at 50–90 mg/kg body-weight. After the mouse was unresponsive to toe pinch, the head was decapitated, and the brain was dissected out and immediately kept in a beaker with ice-cold PBS. The fresh brain proceeded to be cut into 250-300 μm thick coronal slices using Leica Vibratome (VT1000 S). Fresh brain was taken out from ice-cold PBS and the cerebellum side was glued to the vibratome cutting platform with VetbondTM. Cutting platform with glued fresh brain was kept in cutting chamber of the vibratome which was filled with slush of ice-cold PBS. Initial coronal sections were trimmed off until the dorsal striatum (with the help of mice brain atlas) was visible and then 2-3 appropriate sections of the dorsal striatum were chosen. Similarly, midbrain sections were identified under a magnifying glass and 2–3 sections were chosen for midbrain regions containing SNc and SNr areas. The dorsal striatum region was carved out using surgical blade from each hemisphere, placed in ice-cold PBS and transferred into 1.5 ml Eppendorf tube. Excess PBS solution was decanted and the Eppendorf tube containing the brain tissue was immediately kept in dry ice. Similarly, the midbrain region from each half of the selected brain slice was carved out under magnifying glass in ice-cold PBS and was kept in a separate 1.5 ml Eppendorf tube. Sometimes same brain areas dissected out from selected 2-3 sections of the same hemisphere were collected into a single Eppendorf tube. The samples were then kept into −80 °C freezer for further processing.

**Biogenic amine analysis using HPLC-ECD**. Biogenic amines were analyzed in the Vanderbilt University Neurochemistry Core as described previously[8]. Briefly, biogenic amine concentrations were determined utilizing an Antec Decade II (oxidation: 0.65) electrochemical detector operated at 33 °C. 20μl samples of the supernatant were injected using a Water 2707 autosampler onto a Phenomenex Kintex C18 HPLC column (100 × 4.60 mm, 2.6 μm). Biogenic amines are eluted with a mobile phase consisting of 89.5% 0.1 M TCA, 10-2 M sodium acetate, 10-4 M EDTA and 10.5 % methanol (pH 3.8). Solvent was delivered at 0.6 ml/min using a Waters 515 HPLC pump. Using this HPLC solvent the following biogenic amines were eluted in the following order: Norepinephrine, Epinephrine, DOPAC, Dopamine, 5-HIAA, HVA, 5-HT, and 3-MT. Data acquisition was managed by Empower software (Waters Corporation, Milford, MA USA). Isoproterenol (5 ng/mL) was included in the homogenization buffer for use as an internal standard to quantify the biogenic amines of interest.

**Statistical analyses**. All the data were analyzed by Prism 8 software (Graphpad). Data is presented as mean ± SEM or mean ± SD. N represents animal numbers and is indicated in the figure legends. Statistical significance was determined by comparing means of different groups using $t$ test or ANOVA followed by post hoc tests.

**Reporting summary**. Further information on research design is available in the Nature Research Reporting Summary linked to this article.

## Data availability

The accession number of the previously published striatal tissue RNA-seq is PRJNA612478. The accession number of SNc RNA-seq data is PRJNA775656. The source data are provided with this paper as a Source Data file. Source data are provided with this paper.

## Code availability

A linear unmixing algorithm (https://www.niehs.nih.gov/research/atniehs/labs/ln/pi/iv/tools/index.cfm) is available to quantify the emission spectra of eCB2.0, DA2m, GCaMP6f and tdTomato signals captured by fiber photometry.

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

## Acknowledgements

This work is supported in part by the National Key Plan for Scientific Research and Development of China grants (BT, Grant No. 2016YFC1306000), the National Natural Science Foundation of China (BT, Grant No. 81430023), and the Intramural Research Programs of National Institute on Aging, NIH (HC, ZIA AG000944, AG000928), National Institute of Alcoholism and Alcohol Abuse (DML, ZIA AA000416; AS, K99/R00 AA025991), and National Institute of Environmental Health Sciences (GC, ZIA ES103310). We are indebted to the participation of the patients and their family members in this study. We thank Accelerating Medicine Partnership- Parkinson's Disease (AMP-PD, https://www.amp-pd.org) Knowledge Platform for allowing us to use the datasets from European decent. More detailed AMP-PD and AMP-PD Cohort acknowledgements are included in the supporting information. We thank NIMH rodent behavioral core for assisting in behavioral tests, Dr. Josephine M. Egan of NIA for providing the *Cnr1*loxP/loxP mice, Dr. Ku-Lung Hsu from University of Virginia for providing the *Daglb* germline KO mice, and members of Cai lab for their suggestions and technical assistance. Drs. Zhenhua Liu and Nannan Yang were participants in the NIH Graduate Partnership Program and graduate students at Central South University. Dr. Wotu Tian was a participant in the NIH Graduate Partnership Program and a graduate student at Shanghai Jiao Tong University School of Medicine. Dr. Jie Dong was a participant in the NIH Graduate Partnership Program and a graduate student at Dalian Medical University.

## Author contributions

B.T. designed and supervised the human genetics study. H.C. conceived, designed, and supervised the mouse experiments. H.C. wrote the manuscript and prepared the figures with inputs from all authors. Z.L. performed human genetics study, biochemistry, histology, and mouse behavioral experiments, and prepared figures and tables of genetics data. N.Y. and J.D. performed main fiber photometry experiments and data analyses. W.T. generated and analyzed the CB1 conditional KO mice and performed the rescue experiments. J.G., J.M., P.C., and T.W. contributed to human genetic study in China. J.T. and Z. Z. contributed to the biochemistry study. W.T., S.C., S.H., J.K., J.W., M.K., B.T.S., B.B.O., and W.L. contributed to additional behavior tests, histology, and data analyses. A.G.S., D.M.L., L.W., J.Z., and G.C. contributed to additional fiber photometry, histology, and data analyses. L.C. performed stereotaxic surgery. L.S., C.X., and J.H.D. contributed to RNA-sequencing and data analyses. Y.L., A.D., and K.H. provided eCB2.0 and DA2m sensors. R.C. performed endocannabinoids Mass spec measurements. V.P., M.B.M., C.B., and A.B.S. conducted AMP-PD search. All authors read and approved the final manuscript.

## Funding

## Competing interests

The authors declare no competing interests.

## Additional information

[1]Transgenic Section, Laboratory of Neurogenetics, National Institute on Aging, National Institutes of Health, Bethesda, MD 20892, USA. [2]Department of Neurology, Xiangya Hospital, Central South University, 410008 Changsha, Hunan, China. [3]Clinical Research Center on Neurological Diseases, the First Affiliated Hospital, Dalian Medical University, 116011 Dalian, Liaoning, China. [4]Department of Neurology, Ruijin Hospital Affiliated to Shanghai Jiao Tong University School of Medicine, 20025 Shanghai, China. [5]Department of Neurology, Xuanwu Hospital of Capital Medical University, 100053 Beijing, China. [6]Centre for Medical Genetics and Hunan Key Laboratory of Medical Genetics, School of Life Sciences, Central South University, 410008 Changsha, Hunan, China. [7]State Key Laboratory of Membrane Biology, Peking University School of Life Sciences, 100871 Beijing, China. [8]PKU-IDG/McGovern Institute for Brain Research, 100871 Beijing, China. [9]Peking-Tsinghua Center for Life Sciences, Academy for Advanced Interdisciplinary Studies, Peking University, 100871 Beijing, China. [10]In Vivo Neurobiology Group, Neurobiology Laboratory, National Institute of Environmental Health Sciences, Research Triangle Park, NC 27709, USA. [11]Laboratory of Physiologic Studies, National Institute on Alcohol Abuse and Alcoholism, National Institutes of Health, Bethesda, MD 20892, USA. [12]Laboratory for Integrative Neuroscience, National Institute on Alcohol Abuse and Alcoholism, National Institutes of Health, Rockville, MD 20852, USA. [13]Molecular Genetics Section, Laboratory of Neurogenetics, National Institute on Aging, National Institutes of Health, Bethesda, MD 20892, USA. [14]Computational Biology Group, Laboratory of Neurogenetics, National Institute on Aging, National Institutes of Health, Bethesda, MD 20892, USA. [15]Department of Neurology, Union Hospital, Tongji Medical College, Huazhong University of Science and Technology, 430022 Wuhan, Hubei, China. [16]Department of Neurosciences, University of South China Medical School, 421200 Hengyang, Hunan, China. [17]Institute of Neurology, Sichuan Academy of Medical Sciences-Sichuan Provincial Hospital, Medical School of University of Electronics & Technology of China, 610045 Chengdu, Sichuan, China. [18]Integrative Neurogenomics Unit, Laboratory of Neurogenetics, National Institute on Aging, National Institutes of Health, Bethesda, MD 20892, USA. [19]Center for Alzheimer's and Related Dementias, National Institutes of Health, Bethesda, MD 20892, USA. [20]Chinese Institute for Brain Research, 102206 Beijing, China. [21]National Clinical Research Center for Geriatric Disorders, Xiangya Hospital, Central South University, 410008 Changsha, Hunan, China. [22]Key Laboratory of Hunan Province in Neurodegenerative Disorders, Central South University, 410008 Changsha, Hunan, China. [23]These authors contributed equally: Zhenhua Liu, Nannan Yang, Jie Dong. [24]These authors jointly supervised this work: Huaibin Cai, Beisha Tang. ✉email: caih@mail.nih.gov; bstang7398@163.com

