## [Peer Review File · Nature Communications]

Reviewers' Comments:

Reviewer #1:

Remarks to the Author:

This study implicates endocannabinoid (eCB) signaling in regulating the activity of midbrain dopamine (DA) neurons and in the pathophysiology and symptomatic treatment of Parkinson's disease (PD). To do so, Liu and colleagues first identify mutations in DAGLB – a gene whose protein product mediates the biosynthesis of the eCB 2-AG – in four families out of a large cohort of patients with autosomal recessive PD. These mutations localize to different regions of the DAGLB gene, but the authors establish that they all result in loss of function by disrupting DAGLB protein translation and/or stability. They then show that DAGLB mRNA is expressed in 0.5% of SNc DA neurons in humans and 6% of SNc DA neurons in mice, but fail to detect DAGLB protein in mouse SNc DA neurons using commercial antibodies. The authors proceed to demonstrate that systemic administration of a drug that blocks 2-AG degradation (JZL184) elevates eCB levels in the substantia nigra (SN) region using photometry of a novel eCB fluorescent sensor, and that this elevation is reduced in mice in which DAGLB is knocked down (KD) in SNc DA neurons using CRISPR/Cas9, implicating DA neurons in the production of 2-AG in the ventral midbrain of mice. Next, the authors discover that within-session eCB fluorescence intensity in SN on a rotarod motor learning task correlates with learning and that learning is impaired in mice with DAGLB KD in DA neurons (but not in DAGLB knockout animals). In addition, they show that boosting 2-AG production with JZL184 elevates the activity of midbrain DA neurons and DA release in striatum and that JZL184 can restore motor deficits in mice with DAGLB KD in DA neurons. Overall, I found this study to be interesting, as it points to eCBs as a relatively unexplored contribution to SNc DA neuron activity and PD, but my enthusiasm is severely limited by important technical and conceptual concerns that call for major revisions and possibly re-interpretation of the data.

My main concern relates to the claim that DAGLB expressed in SNc DA neurons is the main contributor of 2-AG produced in ventral midbrain and an important contributor to SNc DA neuron activity and DA release. The evidence substantiating this claim mainly stems from the observation that SNc DA neurons contain more DAGLB mRNA than DAGLA, and from a single manipulation (knocking down DAGLB in midbrain DA neurons using viral delivery of Cas9 and CRISPR guide RNA). However, important controls are missing and a few observations raise concerns as to the validity of the proposed model:

- 1) DAGLB mRNA is expressed in an exceedingly small fraction of SNc DA neurons in humans (0.5%) and mice (6%). It is difficult to imagine how so few cells contribute to the bulk of the 2-AG detected by photometry in SN. Do other cells in ventral midbrain or axons projecting into the ventral midbrain express DAGLB?
- 2) The authors state that commercial antibodies do not detect DAGLB in SNc DA neurons. This could be because these antibodies are not good, as the authors suggest, or because DAGLB mRNA is actually not translated into protein in DA neurons. The authors need to distinguish these possibilities by demonstrating that the antibodies are specific for DAGLB in knockout mice, which the authors possess. If specific, immunofluorescence results need to be re-interpreted to indicate that DAGLB protein is not abundant in DA neurons.
- 3) The authors use a AAV-mediated CRISPR approach to knock down DAGLB from DA neurons. They report a transduction efficiency of 75% but do not report on the resulting prevalence of DAGLB-expressing DA neurons: Are they reduced in number by 75% too, bringing mice close to the prevalence found in humans? Or are DAGLB-expressing DA neurons more or less likely to be transduced by this viral construct? In addition, the authors do not comment on specificity: is Cas9 only found in DAT-positive neurons in SNc, or is some expression non-specific? Because AAVs are known to travel retrogradely, the authors need to verify the absence of DAT-positive afferent neurons (in locus coeruleus, for example) that may contribute to the reported effects. These characterizations are essential to evaluate the specificity of this method and to interpret the data presented.
- 4) The authors mention that DAGLB knockout mice do not develop PD-related neuropathological and behavioral abnormalities (including motor learning on rotarod). They conclude that these mice are not a good model to study eCB signaling in motor behavior, favoring instead CRISPR/Cas9 KD. Although compensatory mechanisms in knockout mice are possible, an alternative hypothesis is that DAGLB is not an important contributor to 2-AG in ventral midbrain. Are eCB levels in SN different in knockout mice vs. controls during rotarod learning? This experiment is a must to

support the claim that 1) DAGLB is the main contributor of 2-AG in SN, and 2) that diminished 2-AG production in SN compromises motor learning.

5) Given the centrality of their CRISPR KD manipulation for their conclusions, the authors need to rule out the possibility that it has off-target effects that compromise 2-AG signaling and/or motor behavior. At minimum, a better justification for placing greater trust in KD phenotypes over the knockout is warranted. I note here that the KD model is unlikely to model the autosomal recessive PD mediated DAGLB mutations described in Fig. 1, as these loss-of-function mutations are present at birth, and therefore ought to be better modeled in knockout mice. The authors ought to discuss this caveat.

My second major concern relates to the reported relationship between eCB signal measured by photometry and motor skill learning and/or performance:

1) Fig 4B shows that eCB levels increase a lot between first and last trial in the first session, but not on the 6th. This could reflect, as the authors suggest, that eCB is no longer produced over the course of successive trials, stunting additional learning. Alternatively, the same effect may be mediated by rising baseline eCB levels between days 1 and 6 (since baseline fluorescence (Fb) is used to calculate $\Delta F/Fb$). Can the authors exclude this possibility?

2) In Fig 4E, the authors report a high correlation coefficient between eCB signal and rotarod performance, but methods are not provided. How was this calculated? Is this correlation mediated by the fact that mean eCB signal and motor performance typically increase within each session? If moment-to-moment performance is indeed controlled by eCB signaling, correlations should instead be calculated on a trial by trial basis.

3) Figs 5A-B repeat eCB measurements during rotarod learning from Fig 4, but the relationship is a lot less striking here. There is for instance very little learning on day 3 despite strong increases in eCB levels, and similar within-session learning on days 1 and 6 despite different eCB increases. In addition, eCB signals typically behave comparably during the first 5 trials of each session and deviate from one another during trials 6-10; the behavior does not reflect this, as performance increases similarly within session for each of the two groups. The main difference appears to be in the maintenance of motor learning between days, which the current study does not correlate with eCB levels.

Lastly, the therapeutic promise of JZL184 for PD and the underlying mechanisms presented in Figures 6 and 7 are not clear. The authors show that elevating 2-AG with JZL184 increases DA neuron activity in SNc and DA release in striatum, but that both effects require DAGLB in DA neurons. However, in Figure 7, the authors show that the motor learning/performance deficits of DAGLB KD mice are reversed with JZL184 (mice actually perform better than controls). How do the authors envision this working if, as shown in Figures 3I and 6C/F, knocking down DAGLB curbs the increase in 2-AG and DA evoked by JZL184? In addition, translationally, how do the authors envision JZL184 providing therapeutic relief when the DA neurons that supposedly express DAGLB and produce 2-AG have degenerated? In order for JZL184 to be considered as a PD treatment, the authors have to use a model of PD with neurodegeneration, but their current model suggests that JZL184 ought to no longer be effective when DAGLB-expressing DA neurons are lost.

Reviewer #2:

Remarks to the Author:

The authors report four novel loss-of-function mutations in DAGLB linked to early-onset Parkinson's disease (EOPD). They further demonstrated that DAGLB is the dominant 2-2-arachidonoyl-glycerol (AG) synthase in nigral dopaminergic neurons. Genetic knockdown of *Daglb* in mouse nigral dopaminergic neurons resulted in reduced nigral 2-AG levels and impaired motor skill learning, whereas pharmacological inhibition of 2-AG degradation increased nigral 2-AG levels, promoted dopamine release, and rescued motor deficits.

The results of this study are novel, interesting and potentially relevant for our understanding of Parkinson's disease (PD). The experimental design is elegant and comprehensive and the manuscript is overall well written.

I have got a few comments and questions, some major and some minor:

- The authors should quote a more recent review article on PD (the one they chose is 4 years old) and for monogenic PD, please quote a review that actually addresses this topic (Nalls et al. is a beautiful study and paper but focuses on complex genetics of PD).
- It is not clear whether Family 1 was part of the original sample of 65 families (or collected later)? What is meant by "the remaining ARPD families". If this family was part of the original dataset, why was this mutation not reported before?
- Could the authors please list any potential pathogenic variants in the other four regions of homozygosity? How did they choose to focus on this one region?
- Is there any clinical data on Family 3?
- Please provide coverage for the exam data at 30x (coverage was supposed to be 100x according to the methods).
- How did the authors assess (and exclude) overlapping compound-heterozygous variants (FLG)?
- Please provide the actual CADD scores.
- Were the heterozygous carriers examined by a movement disorder specialist - were there any (subtle) clinical signs in heterozygotes?
- A comparison (and especially stating a difference to) patients with PRKN, PINK1, DJ-1 pathogenic variants is not possible given the very small number of affected with DAGLB mutations (who definitely overlap with the spectrum of the other three genes).
- The authors should comment on the fact that DAGLB pathogenic variants appear to be exceedingly rare (4 index cases among 1500 EOPD patients).
- They should screen other available databases for pathogenic variants in this gene (eg, AMP-PD) which will also include ethnicities other than Chinese. This would be important to better understand the potential significance of their finding in different populations and ethnicities.

Reviewer #3:

Remarks to the Author:

Endocannabinoids are regulators of synaptic functions in the brain, and they can suppress neurotransmitter release both temporarily or in a long-lasting manner. Their involvement in neurodegenerative disorders, such as Alzheimer's disease, Huntington's disease, and Parkinson's disease, has been studied for years. The findings suggest that there is a direct link between endocannabinoid signaling and PD symptoms, and that modulation of this signaling might prove useful for relieving those symptoms. However, the exact mechanism how endocannabinoids cause PD symptoms has not been investigated. In this work, the authors have identified an autosomal recessive mutation in one of the endocannabinoid synthesizing enzymes, DALGB, in some PD families. They show that Dalgb is expressed in both human and mouse DA neurons, and that its targeted inactivation in the adult mouse DA neurons by Dat-Cre-activated AAV-Crispr-Cas9 targeting system leads to specific defects in motor learning without affecting DA neuron viability. Although, as the authors note, the inactivation of Dalgb in the adult DA neurons using a floxed Dalgb and Dat-CreERT2, or a similar system, would provide the most complete inactivation, the phenotype seen with viral knock-down is already displaying a distinct phenotype. Together these results provide a thorough examination on the role of DALGB and endocannabinoids in the dopaminergic function, with implications on PD, and with some additions and clarifications (see comments below), especially regarding the effect of inflammation, the work should be considered for publication in Nature Communications.

Major comments:

-Endocannabinoids possess anti-inflammatory properties (e.g. Turcotte et al., *Journal of Leukocyte Biology* 2015), and inflammation in turn might have a negative effect on dopamine release and motor learning (Felger and Treadway, *Neuropsychopharmacology reviews* 2016). Although loss of *Dalgb* by itself might not induce inflammation, the lack of it might lead to prolonged inflammation following stereotactic injection of viral constructs. The authors should measure the levels of cytokines in the ventral midbrain of DAN *Dalgb*-KD mice, and/or investigate the amounts of activated microglia in that area using histology (e.g. Iba1 antibody).

-Related to the previous comment, concentrations of not only dopamine, but also norepinephrine, DOPAC, 5HIAA, HVA, and 5HT in the DAN *Dalgb* knock-down and control brains should be measured in normal conditions using standard HPLC methods.

Minor comments:

-Although the number of *Dalgb*-deficient DA neurons is unaltered, even in 12-month old mice, authors should investigate whether these neurons show any milder structural changes, for example, altered mitochondrial microstructure or axonal swellings.

-In lines 380-381 the authors write: "-- *Dalgb*, *Parkin*, *Dj-1*, and *Pink1* germline KO mice all failed to develop any PD-like behavioral and pathological phenotypes." However, this statement is not entirely true, as aged (120-week old) *Parkin* mutant mice do display defects in their DA neurons (Noda et al., *Neurobiology of Disease*, 2020), and the wording of that sentence should be adjusted accordingly.

REVIEWER COMMENTS

Reviewer #1 (Remarks to the Author):

This study implicates endocannabinoid (eCB) signaling in regulating the activity of midbrain dopamine (DA) neurons and in the pathophysiology and symptomatic treatment of Parkinson's disease (PD). To do so, Liu and colleagues first identify mutations in DAGLB – a gene whose protein product mediates the biosynthesis of the eCB 2-AG – in four families out of a large cohort of patients with autosomal recessive PD. These mutations localize to different regions of the DAGLB gene, but the authors establish that they all result in loss of function by disrupting DAGLB protein translation and/or stability. They then show that DAGLB mRNA is expressed in 0.5% of SNc DA neurons in humans and 6% of SNc DA neurons in mice but fail to detect DAGLB protein in mouse SNc DA neurons using commercial antibodies. The authors proceed to demonstrate that systemic administration of a drug that blocks 2-AG degradation (JZL184) elevates eCB levels in the substantia nigra (SN) region using photometry of a novel eCB fluorescent sensor, and that this elevation is reduced in mice in which DAGLB is knocked down (KD) in SNc DA neurons using CRISPR/Cas9, implicating DA neurons in the production of 2-AG in the ventral midbrain of mice. Next, the authors discover that within-session eCB fluorescence intensity in SN on a rotarod motor learning task correlates with learning and that learning is impaired in mice with DAGLB KD in DA neurons (but not in DAGLB knockout animals). In addition, they show that boosting 2-AG production with JZL184 elevates the activity of midbrain DA neurons and DA release in striatum and that JZL184 can restore motor deficits in mice with DAGLB KD in DA neurons. Overall, I found this study to be interesting, as it points to eCBs as relatively unexplored contribution to SNc DA neuron activity and PD, but my enthusiasm is severely limited by important technical and conceptual concerns that call for major revisions and possibly re-interpretation of the data.

RESPONSE: We appreciate the reviewer's general support of this study.

My main concern relates to the claim that DAGLB expressed in SNc DA neurons is the main contributor of 2-AG produced in ventral midbrain and an important contributor to SNc DA neuron activity and DA release. The evidence substantiating this claim mainly stems from the observation that SNc DA neurons contain more DAGLB mRNA than DAGLA, and from a single manipulation (knocking down DAGLB in midbrain DA neurons using viral delivery of Cas9 and CRISPR guide RNA). However, important controls are missing, and a few observations raise concerns as to the validity of the proposed model:

1) DAGLB mRNA is expressed in an exceedingly small fraction of SNc DA neurons in humans (0.5%) and mice (6%). It is difficult to imagine how so few cells contribute to the bulk of the 2-AG detected by photometry in SN. Do other cells in ventral midbrain or axons projecting into the ventral midbrain express DAGLB?

RESPONSE: Here we are afraid that the reviewer had misread the data presented in **Fig. 2A-C**. The data are **not** about the percentage of TH-positive neurons that express *DAGLB* mRNA. As shown in **Fig. 2D** with RNAscope *in situ* hybridization, *Daglb* mRNA is widely expressed by both TH-positive and -negative neurons in the SNc. Since the ratios of TH-positive neurons were varied in each isolated midbrain samples, to normalize the expression level of *DAGLB* mRNA in dopaminergic neurons, we calculated the relative expression levels of *DAGLB* mRNA in each sample as the percentage of *TH* mRNA levels. Therefore, in human SNc dopaminergic neurons,

the average *DAGLB* mRNA level is about 0.5% of TH mRNA level (**Fig. 2A**), while in mouse SNc dopaminergic neurons the percentage is about 6% (**Fig. 2B**). To avoid any confusion, in the revised manuscript, we labeled the Y-axis as “% of TH mRNA” in **Fig. 2A**, “% of *Th* mRNA” in **Fig. 2B**, and “% of *Bcl11b* mRNA” in **Fig. 2C**.

2) The authors state that commercial antibodies do not detect DAGLB in SNc DA neurons. This could be because these antibodies are not good, as the authors suggest, or because DAGLB mRNA is not translated into protein in DA neurons. The authors need to distinguish these possibilities by demonstrating that the antibodies are specific for DAGLB in knockout mice, which the authors possess. If specific, immunofluorescence results need to be re-interpreted to indicate that DAGLB protein is not abundant in DA neurons.

RESPONSE: In **Fig. 3B**, we showed in western blot that the level of DAGLB protein was substantially reduced in the *Daglb* knock-down (KD) sample, demonstrating the specificity of DAGLB antibody. To further support this notion, we repeated the western blot in the *Daglb* knockout (KO) mouse brains as suggested by the reviewer and showed a complete loss of DAGLB protein expression in the cortex, striatum, and cerebellum of *Daglb* KO mice (**Fig. 1**). Together, these results demonstrate the specificity of DAGLB antibody. Many factors could contribute to the difficulty in immunostaining. The relatively low expression level of DAGLB protein in midbrain dopaminergic neurons could be a factor. Additionally, compared to the western blot, in which the proteins were denatured and unfolded, the DAGLB proteins may exist as the native folded conformation in tissue sections, which could potentially mask the antibody binding sites and reduce the accessibility of antibody. We added these notions on Page 23 of the revised Discussion section.

Fig. 1 Western blots of DAGLB and CB1 expression in the cortex (CX), striatum (ST), and cerebellum (CB) of wild-type (WT) and *Daglb* KO (KO) mice.

3) The authors use an AAV-mediated CRISPR approach to knock down DAGLB from DA neurons. They report a transduction efficiency of 75% but do not report on the resulting prevalence of DAGLB-expressing DA neurons: Are they reduced in number by 75% too, bringing mice close to the prevalence found in humans? Or are DAGLB-expressing DA neurons more or less likely to be transduced by this viral construct? In addition, the authors do not comment on specificity: is Cas9 only found in DAT-positive neurons in SNc, or is some

expression non-specific? Because AAVs are known to travel retrogradely, the authors need to verify the absence of DAT-positive afferent neurons (in locus coeruleus, for example) that may contribute to the reported effects. These characterizations are essential to evaluate the specificity of this method and to interpret the data presented.

RESPONSE: Our AAV-mediated CRISPR approach to knock down *Daglb* from nigral dopaminergic neurons is both specific and efficient. In **Fig. 3B**, we demonstrated that the AAV-*Daglb* KD gene targeting vectors successfully disrupted the DAGLB protein expression in cultured neurons by western blot. The saCas9-mediated genomic modification should be permanent, resulting in a complete knockout of *Daglb* gene expression in the saCas9-sg*Daglb* expressing cells. In strong support of the genetic deletion, the fiber photometry and mass spectrometry assays demonstrated a substantial reduction of 2-AG levels in the KD mice (**Fig. 3E-K**). We believe that the reviewer's comment about prevalence is due to a misunderstanding that only a small fraction of TH-positive neurons express DAGLB. As we discussed earlier, DAGLB is widely expressed by TH-positive neurons in the SNc.

In the supplementary **Fig. S10C**, we estimated that an average of 75% of SNc TH-positive neurons were transduced by the viral vectors in each animal following the stereotactic injections. Since the expression of saCas9 is Cre dependent (**Fig. 3A**) and the viral vectors were stereotactically injected in the SNc of DAT^{IRESCre} mice (**Fig. S10A**), the saCas9 was mainly expressed by the SNc TH-positive neurons as demonstrated in **Fig. S10B**. We used HA staining to visualize the distribution of HA-tagged saCas9 in different brain regions. Except for the midbrain, we didn't observe any HA staining in other brain regions, including *locus coeruleus* (LC). The AAV9 serotype possesses very limited capacity in retrograde axonal transport¹. However, even if the AAV9 vector could effectively get into the axons, none of those SNc-projecting neurons express DAT. For example, the LC norepinephrinergic neurons express TH but not DAT. DAT is only expressed by the dopaminergic neurons. Our observations are in line with the initial characterization of DAT^{IRESCre} mice, in which the Cre-dependent gene expression is only observed in the midbrain dopaminergic neurons and to a lesser extent in the olfactory glomeruli². Those olfactory dopaminergic interneurons do not extend their axons outside of olfactory bulb. Therefore, with the combination of DAT^{IRESCre} mice and local infusion of Cre-dependent viral vectors in the SNc, the spread of *Daglb* knock-down outside of midbrain dopaminergic neurons is not a realistic concern. We included the above discussion on Page 22 of the revised Discussion section.

4) The authors mention that DAGLB knockout mice do not develop PD-related neuropathological and behavioral abnormalities (including motor learning on rotarod). They conclude that these mice are not a good model to study eCB signaling in motor behavior, favoring instead CRISPR/Cas9 KD. Although compensatory mechanisms in knockout mice are possible, an alternative hypothesis is that DAGLB is not an important contributor to 2-AG in ventral midbrain. Are eCB levels in SN different in knockout mice vs. controls during rotarod learning? This experiment is a must to support the claim that 1) DAGLB is the main contributor of 2-AG in SN, and 2) that diminished 2-AG production in SN compromises motor learning. 5) Given the centrality of their CRISPR KD manipulation for their conclusions, the authors need to rule out the possibility that it has off-target effects that compromise 2-AG signaling and/or motor behavior. At minimum, a better justification for placing greater trust in KD phenotypes over the knockout is warranted. I note here that the KD model is unlikely to model the autosomal

recessive PD mediated DAGLB mutations described in Fig. 1, as these loss-of-function mutations are present at birth, and therefore ought to be better modeled in knockout mice. The authors ought to discuss this caveat.

RESPONSE: We agree with the reviewer that our present mouse study was not intended to model the disease per se, rather to investigate the role of *DAGLB*-mediated 2-AG biothesis in SNc dopaminergic neurons in regulating motor behaviors. Compared to the germline *Daglb* KO mice, the saCas9-mediated gene targeting strategy used in the present study allowed for specifically evaluating the impact of *DAGLB*-deficiency on adult SNc dopaminergic neurons in a cell type-specific manner. In **Fig. 3I-K**, we showed that genetic knock-down of *Daglb* in nigral dopaminergic neurons led to substantial reduction of 2-AG levels in the SN region by both fiber-photometry and mass-spec measurements, demonstrating that *Daglb*-mediated 2-AG biosynthesis in nigral dopaminergic neurons contributes substantially to the overall production of 2-AG in the SN. However, the germline *Daglb* KO mice is not a suitable model to study the role of *Daglb* specifically in nigral dopaminergic neurons. Besides nigral dopaminergic neurons, *DAGLB* is also expressed in other brain regions involved in motor learning, including striatum, cortex, and cerebellum (**Fig. 1**). Thus, it is likely that the germline *Daglb* KO could induce complex compensatory responses to 2-AG deficiency, as well as circuit plasticity during development to retain normal motor learning. Since the germline *Daglb* KO mice did not show any impairments in rotarod tests at both 4 and 20 months of age (**Fig. S14**), it would be difficult to link 2-AG levels to rotarod performance in those mice. Even if it is possible that a similar reduction of 2-AG levels might be observed in the SN of germline *Daglb* KO mice during rotarod tests, it would be difficult to attribute the change of 2-AG levels to the dopaminergic neurons since many non-dopaminergic neurons in SNc and SNr also express *DAGLB* and contribute to the biosynthesis of 2-AG (**Figs. 2D, S9**). Therefore, the additional fiber photometry recording of 2-AG levels in the KO mice may not be particularly informative compared to the current studies with the *Daglb*-KD mice.

The reviewer raised a valid concern regarding potential off-target effect of the *Daglb*-KD gene targeting vector. While it might be difficult to completely rule out the off-target effect of any CRISPR/Cas9 gene targeting vector, our western blot and fiber photometry data strongly support the specificity of the *Daglb*-KD gene targeting vector used in the current study. The sgRNA was chosen based on the unique sequence in the 3'-untranslated region of *Daglb* gene. Accordingly, in **Fig. 3B**, we demonstrated by western blot that the AAV-*Daglb* KD gene targeting vectors specifically disrupted the expression of *DAGLB* but not the closely related *DAGLA* expression in cultured neurons. In **Figs. 3I** and **S11**, we showed by fiber photometry that the knock-down of *Daglb* but not *Dagla* in the SNc dopaminergic neurons led to significant reduction of 2-AG levels in the SN compared to the control saCas9-empty vectors. The negative results from the saCas9-empty and *Dagla* sgRNA viral vectors strongly argue against the possibility that any of our findings in the *Daglb* KD mice are due to some off-target effect of gene targeting vectors.

My second major concern relates to the reported relationship between eCB signal measured by photometry and motor skill learning and/or performance:

1) *Fig 4B shows that eCB levels increase a lot between first and last trial in the first session, but not on the 6th. This could reflect, as the authors suggest, that eCB is no longer produced over the course of successive trials, stunting additional learning. Alternatively, the same effect may be*

mediated by rising baseline eCB levels between days 1 and 6 (since baseline fluorescence (Fb) is used to calculate $\Delta F/Fb$). Can the authors exclude this possibility?

RESPONSE: The reviewer raised an interesting possibility. However, as a matter of fact, fiber photometry measures the relative change of fluorescence levels against a selected baseline level. Such baseline fluorescence (Fb) levels were fluctuated, going either up or down in the same animal when we set up the fiber photometry recording in the beginning of any given day (i.e., the initial gross baseline level varied irregularly between days). Many factors could contribute to such variations, such as the reconnection of implanted optic fiber to the fiber photometry recording device on different days. Therefore, simple comparison of raw baseline eCB levels (Fb) measured in photometry across recording days will not reflect actual changes of eCB levels. To overcome this day-to-day variability, we calculated the $\Delta F/Fb$ to normalize the changes of fluorescent signals daily.

2) In Fig 4E, the authors report a high correlation coefficient between eCB signal and rotarod performance, but methods are not provided. How was this calculated? Is this correlation mediated by the fact that mean eCB signal and motor performance typically increase within each session? If moment-to-moment performance is indeed controlled by eCB signaling, correlations should instead be calculated on a trial-by-trial basis.

RESPONSE: We thank the reviewer for pointing out this ambiguity in our description. The correlation was calculated on a trial-by-trial basis. For each mouse on a given day, the mean eCB signals in each of the 10 trials and the corresponding rotarod performance (latency to fall) were used to calculate the correlation coefficient (R^2). The average R^2 of eight mice on each day were used to plot the graph in **Fig 4E**. We added the above description on Page 13 of the revised Result section, as well as on Page 40 of the revised Method section.

3) Figs 5A-B repeat eCB measurements during rotarod learning from Fig 4, but the relationship is a lot less striking here. There is for instance very little learning on day 3 despite strong increases in eCB levels, and similar within-session learning on days 1 and 6 despite different eCB increases. In addition, eCB signals typically behave comparably during the first 5 trials of each session and deviate from one another during trials 6-10; the behavior does not reflect this, as performance increases similarly within session for each of the two groups. The main difference appears to be in the maintenance of motor learning between days, which the current study does not correlate with eCB levels.

RESPONSE: We appreciate the efforts made by reviewer for closely cross-examining the eCB signals and corresponding rotarod performance in a trial-by-trial fashion. We agree with the reviewer that the intensities of eCB signals are not strictly correlated with the motor performance especially during trials 6-10 in each session. While the eCB levels continued to build up over sequential trials, the improvement of motor performance apparently was not linear. The complex relationship between eCB transmission and motor performance echoes findings in changes of neuronal ensemble activity in the dorsal striatum during rotarod learning³. This interesting observation likely indicates that the peak rotarod performance is determined at the organism level and by multiple factors. We added the above discussion on Page 23 of revised Discussion section.

We also agree with the reviewer that the eCB signals seems to be important in maintaining the learned motor skills across sessions. We thereby further compared the rotarod performance

between the first trials of each training session, as well as the last trials. The rotarod performance in the first trials was significantly improved across the training sessions in DAN-Ctrl mice (one-way ANOVA, $F=4.344$, $P=0.0014$). The performance on day 3, 5, and 6 was significantly better than day 1 [multiple comparisons, adjusted P value (day 1 vs. day 2-6) = 0.3418, 0.0004, 0.0604, 0.0043, and 0.0103, respectively]. By contrast, there were no significant changes in the rotarod performance in the first trials in DAN-*Daglb* KD mice (one-way ANOVA, $F=0.5854$, $P=0.7111$). The performance on day 2 to 6 was comparable to day 1 [Multiple comparisons, adjusted P value (day 1 vs. day 2-6) = 0.4439, 0.5520, 0.6926, 0.7463 and 0.9633, respectively]. Moreover, the rotarod performance in the first trials on day 3 to 6 was significantly worse in the DAN-*Daglb* KD mice compared to the controls (Multiple t-test, $p=0.0009$, 0.02, 0.0007, and 0.0006, respectively) (revised **Fig. 5C**). In contrary to the first trials in each training session, the rotarod performance in the last trials was largely comparable across the training sessions in DAN-Ctrl mice (one-way ANOVA, $F=2.095$, $P=0.0732$). However, the performance on day 6 was significantly better than day 1 [multiple comparisons, adjusted P value (day 1 vs. day 2-6) = 0.1391, 0.3302, 0.0553, 0.0868, and 0.0215, respectively]. The DAN-*Daglb* KD mice, on the one hand, like the DAN-Ctrl mice, showed no significant changes in the overall rotarod performance in the last trials of each training session (one-way ANOVA, $F=2.177$, $P=0.0636$), although the performance on day 6 was better than day 1 [Multiple comparisons, adjusted P value (day 1 v. day 2-6) = 0.9253, 0.4052, 0.1832, 0.4400, and 0.0138, respectively]. On the other hand, the performance of DAN-*Daglb* KD mice in the last trials on day 2, 4, and 5 was worse than the DAN-Ctrl mice in corresponding training sessions (Multiple t-test, $p=0.0050$, 0.0369, and 0.0191, respectively) (revised **Fig. 5C**). Together, these analyses suggest that *Daglb*-deficiency in nigral DANs particularly affects retention or consolidation of earned motor skills across training sessions. We added these new analyses on Page 15 of revised Results section.

We also conducted further analyses on the rotarod performance in the first and last trials in the *Drd1-Cre/Cnr1^{fl/fl}* (CB1 cKO) mice (revised **Fig. 5F**). The rotarod performance in the first trials was significantly improved across the training sessions in control *Cnr1^{fl/fl}* mice (one-way ANOVA, $F=7.711$, $P<0.0001$). The performance on day 2 to 6 was significantly better than day 1 [multiple comparisons, adjusted P value (day 1 vs. day 2-6) = 0.0229, 0.0010, 0.0002, 0.0001, and <0.0001 , respectively]. Significant improvement in the first trials was also observed in the CB1 cKO mice (one-way ANOVA, $F=5.792$, $P=0.0002$). Additionally, the performance on day 3 to 6 was better than day 1 [Multiple comparisons, adjusted P value (day 1 v. day 2-6) = 0.4999, 0.0035, 0.0088, 0.0036, and <0.0001 , respectively]. However, the performance of CB1 cKO mice in the first trials on day 2, 4, 5, and 6 was worse than the control mice (Multiple t-test, $p=0.0474$, 0.0375, 0.0458 and 0.0204, respectively) (revised **Fig. 5F**). Like in the first trials, while the performance in the last trials of each training session was significantly improved in both control and CB1 cKO mice (one-way ANOVA, *Cnr1^{fl/fl}* mice: $F=4.403$, $P=0.0026$; *Drd1-Cre/Cnr1^{fl/fl}* mice: $F=3.125$, $P=0.0130$), the performance of CB1 cKO mice in the last trials on day 3 to 6 was worse than the control mice (Multiple t-test, $p=0.0232$, 0.0435, 0.0387 and 0.0031, respectively) (revised **Fig. 5F**). Together, these new results suggest that CB1 receptor-deficiency in direct pathway spiny neurons also affects across-session motor skill learning. We added the new analyses and discussion on Page 16 of the revised Results section.

Fig. 5 (C) Box and whiskers plot (min to max) of rotarod performance in the first and 10th trial on each day. Multiple t-test. * $p < 0.05$, ** $p < 0.005$, * $p < 0.001$. (F) Box and whiskers plot (min to max) of rotarod performance in the first and 10th trial on each day. Multiple t-test. * $p < 0.05$, ** $p < 0.005$.**

For the difference in eCB recording in **Fig. 5A** and **Fig. 4**, we need to point out that the data in those two figures were generated from two different mouse strains. Wild-type C657BL6 mice were used in **Fig. 4**, while $\text{DAT}^{\text{IRESCre}}$ mice were used in **Fig. 5A-B**. In addition, the $\text{DAT}^{\text{IRESCre}}$ mice also received local infusion of control or *Daglb*-KD viral vectors in the SNc region. We suspect that different mouse strains and additional surgery for viral infusion may contribute to the difference in the maximal daily increases of eCB signals between the wild-type mice in **Fig. 4** and the control mice in **Fig. 5A**. However, the overall dynamic changes of eCB signals appeared to follow the same trend over the entire 6-day training sessions. More importantly, when compared in the same $\text{DAT}^{\text{IRESCre}}$ mouse strain, the deletion of *Daglb* in SNc dopaminergic neurons led to significant reduction of eCB signals (**Fig. 5A**), demonstrating a critical involvement of DAGLB-mediated 2-AG biosynthesis in the SNc dopaminergic neurons during rotarod motor skill learning.

We also need to point out that two different cohorts of $\text{DAT}^{\text{IRESCre}}$ mice were used in **Fig. 5A** and **Fig. 5B** and they were tested in different rotarod systems. In **Fig. 5B**, a large cohort of 16 mice per genotype were assigned for only the motor behavioral tests, of which the DAN-*Daglb* KD mice showed significant impairments in the rotarod motor skill learning compared to the control DAN-Ctrl mice. In the revised Supplementary **Fig. S13A**, we plotted the rotarod performance of the same cohort of mice used for fiber photometry in **Fig. 5A**. Although the control mice seemed to perform better than the DAN-*Daglb* KD mice, since the mouse numbers were small (5 per genotype), the difference was not statistically significant (2-way ANOVA, genotype: $F_{(1, 8)} = 1.85$, $p = 0.2100$).

Fig. S13A Rotarod motor skill learning performance of the same mouse cohort in Fig. 5A.

4) Lastly, the therapeutic promise of JZL184 for PD and the underlying mechanisms presented in Figures 6 and 7 are not clear. The authors show that elevating 2-AG with JZL184 increases DA neuron activity in SNc and DA release in striatum, but that both effects require DAGLB in DA neurons. However, in Figure 7, the authors show that the motor learning/performance deficits of DAGLB KD mice are reversed with JZL184 (mice actually perform better than controls). How do the authors envision this working if, as shown in Figures 3I and 6C/F, knocking down DAGLB curbs the increase in 2-AG and DA evoked by JZL184? In addition, translationally, how do the authors envision JZL184 providing therapeutic relief when the DA neurons that supposedly express DAGLB and produce 2-AG have degenerated? In order for JZL184 to be considered as a PD treatment, the authors have to use a model of PD with neurodegeneration, but their current model suggests that JZL184 ought to no longer be effective when DAGLB-expressing DA neurons are lost.

RESPONSE: We need to point out that in **Fig. 7**, the treatment of JZL184 led to increase of dopaminergic neural activity and dopamine release in the SNc not striatum. The *DAGLB*-mediated 2-AG production in SNc dopaminergic neurons may enhance the DAN activity and somatodendritic dopamine release through attenuating the inhibitory inputs from direct pathway striatal spiny projection neurons (**a working model in revised Fig. 8B**). Since 2-AG works locally near the production and release sites, we only examined the interplay between 2-AG and dopamine signaling in the SN regions. Future experiments will be performed to investigate whether the change of 2-AG release in the soma and dendrites of nigral DANs affects the dopamine release in DAN axon terminals at dorsal striatum.

Fig. 8B A working model of nigral DAN-derived 2-AG signaling in regulating DAN neuron activity and dopamine release in SN region, where 2-AG modulates DAN calcium influx and somatodendritic dopamine release through retrograde inhibition of presynaptic GABA release from dSPNs via CB1-mediated intracellular signaling pathway. Genetic deletion of *Daglb* in DANs compromises the 2-AG-mediated feedback regulation, while pharmacological inhibition of 2-AG degradation by JZL184 restores the signaling. Our study suggests that this nigral DAN-derived 2-AG signaling in the SN is important in regulating the across-session motor skill learning. DAG: diacylglycerol, DA: dopamine, AA: arachidonic acid.

Although we achieved high efficiency in CRISPR/saCas9-mediated *Daglb* knockdown (~75% of SNc dopaminergic neurons) in the current study, the remaining 25% of SNc dopaminergic neurons still possess intact DAGLB activity and may contribute to the increase of 2-AG levels following JZL184 administration as observed in **Fig. 3I**. The additional sources of 2-AG could come from the production of DAGLA in SNc dopaminergic neurons, as well as both DAGLA and DAGLB from non-dopaminergic neurons in the SNc and SNr. Our findings are consistent with a recent study⁴, in which the presence of JZL184 strongly suppressed the degradation of 2-AG and allowed 2-AG to diffuse for longer distance and act on the neighboring CB1 receptors for longer duration. Therefore, administration of JZL184 in DAN-*Daglb* KD mice resulted higher level of 2-AG in SN comparing to vehicle treated controls (**Fig. 3J**), which may contribute to better motor learning/performance.

We concur with the reviewer's assessment that the efficacy of JZL184 treatment could be weakened in PD cases with severe dopaminergic neurodegeneration, as our study suggested that the JZL184-induced nigral dopamine release depends on the presence of dopaminergic neurons (**Fig. 7**). Nonetheless, the administration of JZL184 would be beneficial to compensate for the loss of 2-AG signaling in patients who carry the *DAGLB* mutations, as well as to enhance dopamine release in patients who remain to have enough dopaminergic neurons. We added the above notions on Page 22 and 25 of the revised Discussion section.

Reviewer #2 (Remarks to the Author):

The authors report four novel loss-of-function mutations in DAGLB linked to early-onset Parkinson's disease (EOPD). They further demonstrated that DAGLB is the dominant 2-2-arachidonoyl-glycerol (AG) synthase in nigral dopaminergic neurons. Genetic knockdown of Daglb in mouse nigral dopaminergic neurons resulted in reduced nigral 2-AG levels and impaired motor skill learning, whereas pharmacological inhibition of 2-AG degradation increased nigral 2-AG levels, promoted dopamine release, and rescued motor deficits.

The results of this study are novel, interesting and potentially relevant for our understanding of Parkinson's disease (PD). The experimental design is elegant and comprehensive, and the manuscript is overall well written.

RESPONSE: We appreciate the very positive feedback from the reviewer.

I have got a few comments and questions, some major and some minor:

- The authors should quote a more recent review article on PD (the one they chose is 4 years old) and for monogenic PD, please quote a review that actually addresses this topic (Nalls et al. is a beautiful study and paper but focuses on complex genetics of PD).

RESPONSE: As suggested by the reviewer, we replaced the references with two recent review articles on the PD genetics^{5,6}, including the monogenetic PD.

- It is not clear whether Family 1 was part of the original sample of 65 families (or collected later)? What is meant by "the remaining ARPD families"? If this family was part of the original dataset, why was this mutation not reported before?

RESPONSE: The Family 1 is not part of the original 65 families reported previously in *Brain*⁷.

Those 65 families carry 23 known PD-related recessive mutations. Since the mutation had yet to be identified in Family 1 at that time, we didn't report the family in the paper.

- *Could the authors please list any potential pathogenic variants in the other four regions of homozygosity? How did they choose to focus on this one region?*

RESPONSE: We checked the other four regions in the two affected individuals but didn't find any potential pathogenic homozygous variants other than *DAGLB*.

- *Is there any clinical data on Family 3?*

RESPONSE: Yes, we provided the clinical data in the supplementary "Clinical Description" and "Table S4".

- *Please provide coverage for the exam data at 30x (coverage was supposed to be 100x according to the methods).*

RESPONSE: Yes, we added the coverage for the exam data at 30x in the revised supplementary "Table S2".

- *How did the authors assess (and exclude) overlapping compound-heterozygous variants (FLG)?*

RESPONSE: We listed the potential compound-heterozygous variants from the initial screening in the last row of Table S2. However, when we examined the original BAM data from the WES, the variants of those genes are in the same chromosome. Therefore, they are not compound-heterozygous variants. We further confirmed the findings with Sanger sequencing of different family members. We thereby removed the row in the revised Table S2.

- *Please provide the actual CADD scores.*

RESPONSE: Yes, we added the CADD scores in the revised Supplementary Table S3.

- *Were the heterozygous carriers examined by a movement disorder specialist - were there any (subtle) clinical signs in heterozygotes?*

RESPONSE: Yes, we examined those individuals with heterozygous mutations but did not observe any clinical signs.

- *A comparison (and especially stating a difference to) patients with PRKN, PINK1, DJ-1 pathogenic variants is not possible given the very small number of affected with DAGLB mutations (who definitely overlap with the spectrum of the other three genes).*

RESPONSE: We agree with reviewer and deleted the sentence in the revised manuscript.

- *The authors should comment on the fact that DAGLB pathogenic variants appear to be exceedingly rare (4 index cases among 1500 EOPD patients).*

RESPONSE: We agree with the reviewer that *DAGLB* variants might not be common genetic variants associated with increase of risk for PD. From the 1742 PD cases, we identified 58 cases with PARKIN, 8 cases with PINK1, 6 cases with PLA2G6, 3 cases with DJ-1, and 2 cases with ATP13A2 pathogenic mutations. Therefore, the occurrence of *DAGLB* pathogenic variants is similar most of the known PD genes, except for PARKIN. On Page 8 of the revised manuscript, we acknowledged that "although exceedingly rare, we linked four different homozygous *DAGLB* mutations to six affected ARPD/EOPD individuals from four ethnic Han Chinese families".

Despite of the rare occurrence, these new *DAGLB* mutations allow us to conduct series of functional studies to demonstrate the pathogenicity of *DAGLB*-deficiency in PD, of which we showed that 1) all the four mutations disrupt the function and stability of *DAGLB* protein, 2) *DAGLB* is the main 2-AG synthase in nigral dopaminergic neurons, 3) genetic deletion of *Daglb* in nigral dopaminergic neurons impairs motor function, and 4) 2-AG augmentation rescues the motor deficits. Therefore, through studying the rare *DAGLB* mutations, we revealed a novel mechanism of *DAGLB*-mediated 2-AG signaling in PD pathogenesis and in regulating the activity of nigral dopaminergic neurons.

- They should screen other available databases for pathogenic variants in this gene (eg, AMP-PD) which will also include ethnicities other than Chinese. This would be important to better understand the potential significance of their finding in different populations and ethnicities.

RESPONSE: As recommended by the reviewer, with the help of Andrew Singleton and his colleagues, we conducted a preliminary search for any potential *DAGLB* pathogenic variants in the Accelerating Medicines Partnership Parkinson's disease (AMP-PD) dataset, which includes 2,556 controls and 1,451 PD cases from unrelated European descent. Using a minor allele frequency (MAF) < 0.05, we identified 30 missense variants and there were no loss-of-function mutations (splicing, stop, frameshift) in the *DAGLB* gene. Following the proposed mechanism of recessive disease, we found 12 out of 1,451 PD cases were recessive or (potentially) compound heterozygous which results in a frequency of 0.008, and 28 out of 2,556 controls were recessive or (potentially) compound heterozygous which results in a frequency of 0.01. We also used SKAT-O, CMC Wald, and CMC burden testing, which resulted in non-significant p-values for all missense variants as defined by ANNOVAR: 1, 0.94, and 0.94 respectively. This analysis suggests that *DAGLB* is not linked to PD cases of European descent. However, one important caveat using AMP-PD data is that these are all "normal" PD cases and not enriched for monogenic or early onset cases as we did in the Chinese PD cases. It may explain why it is always hard to use AMP-PD to replicate rare familial mutations. We included these new results on Pages 8 and 21 of the revised Results and Discussion sections, respectively.

Reviewer #3 (Remarks to the Author):

*Endocannabinoids are regulators of synaptic functions in the brain, and they can suppress neurotransmitter release both temporarily or in a long-lasting manner. Their involvement in neurodegenerative disorders, such as Alzheimer's disease, Huntington's disease, and Parkinson's disease, has been studied for years. The findings suggest that there is a direct link between endocannabinoid signaling and PD symptoms, and that modulation of this signaling might prove useful for relieving those symptoms. However, the exact mechanism how endocannabinoids cause PD symptoms has not been investigated. In this work, the authors have identified an autosomal recessive mutation in one of the endocannabinoids synthesizing enzymes, *DALGB*, in some PD families. They show that *Dalgb* is expressed in both human and mouse DA neurons, and that its targeted inactivation in the adult mouse DA neurons by *Dat*-Cre-activated AAV-Crispr-Cas9 targeting system leads to specific defects in motor learning without affecting DA neuron viability. Although, as the authors note, the inactivation of *Dalgb* in the adult DA neurons using a floxed *Dalgb* and *Dat*-CreERT2, or a similar system, would provide the most complete inactivation, the phenotype seen with viral knock-down is already displaying a distinct phenotype. Together these results provide a thorough examination on the role of *DALGB**

and endocannabinoids in the dopaminergic function, with implications on PD, and with some additions and clarifications (see comments below), especially regarding the effect of inflammation, the work should be considered for publication in *Nature Communications*.

RESPONSE: We appreciate the strong positive feedback and endorsement from the reviewer.

Major comments:

-Endocannabinoids possess anti-inflammatory properties (e.g. Turcotte et al., *Journal of Leukocyte Biology* 2015), and inflammation in turn might have a negative effect on dopamine release and motor learning (Felger and Treadway, *Neuropsychopharmacology reviews* 2016). Although loss of *Daglb* by itself might not induce inflammation, the lack of it might lead to prolonged inflammation following stereotaxic injection of viral constructs. The authors should measure the levels of cytokines in the ventral midbrain of DAN *Daglb*-KD mice, and/or investigate the amounts of activated microglia in that area using histology (e.g. *Iba1* antibody).

RESPONSE: As recommended by the reviewer, we performed additional immunostaining of midbrain sections with *Iba1* antibody. As shown in revised supplementary **Fig. S10F**, the density and morphology of *Iba1*-positive microglia appeared comparable in the midbrain regions of DAN-Ctrl and DAN-*Daglb* KD mice 6-month after stereotaxic surgery. Furthermore, we extracted total RNAs from the midbrains of DAN-Ctrl and DAN-*Daglb* KD mice (n=5 per genotype) and checked the expression of *Th*, *TNF α* , *IL2*, *IL6*, *Ifnb1*, and *Actb* by qRT-PCR. The expression of *Th* and *Actb* was easily detectable; however, we could not detect any expression of *TNF α* , *IL2*, *IL6*, and *Ifnb1* in any of those 10 samples. Together, we don't think the loss of *Daglb* in SNc DANs leads to prolong inflammation following stereotaxic injections.

Fig. S10F Sample images of *Iba1* (green) and TH (red) staining in the SNr of DAN-Ctrl and DAN-*Daglb* KD mice 6-month after stereotaxic surgery. Scale bar: 50 μ m.

-Related to the previous comment, concentrations of not only dopamine, but also norepinephrine, DOPAC, 5HIAA, HVA, and 5HT in the DAN *Daglb* knock-down and control brains should be measured in normal conditions using standard HPLC methods.

RESPONSE: As suggested by the reviewer, we measured the levels of bioamines and their metabolites in the dorsal striatum of DAN-*Daglb* KD and control mice. As shown in revised supplementary **Fig. S10E**, we did not detect any obvious changes between the KD and control mice.

Fig. S10E HPLC quantification of norepinephrine (NE), 3,4-Dihydroxyphenylacetic acid (DOPAC), dopamine (DA), 5-Hydroxyindoleacetic acid (5-HIAA), Homovanillic acid (HVA), 5-hydroxytryptamine (5-HT), and 3-Methoxytyramine (3-MT) in the dorsal striatum of DAN-Daglb KD (n=5) and control mice (n=4). Multiple t-test, no significant group difference.

Minor comments:

*-Although the number of *Daglb*-deficient DA neurons is unaltered, even in 12-month-old mice, authors should investigate whether these neurons show any milder structural changes, for example, altered mitochondrial microstructure or axonal swellings.*

RESPONSE: As suggested by the reviewer, we stained the striatum and midbrain sections of 12-month-old DAN-Ctrl and DAN-Daglb KD mice with antibodies against TH and TOM20 for the mitochondrial and axonal morphology, respectively in revised supplementary **Fig. S10G, H**. As shown in the sample images, we did not observe any apparent structural changes in the KD samples.

Fig. S10 (G) Sample images of mitochondrial marker TOM20 (green) and TH (red) staining in the SNc of DAN-Ctrl and DAN-Daglb KD mice at 12 months of age. Scale bar: 20µm. **(H)** Sample images of TH staining in the dorsal striatum of DAN-Ctrl and DAN-Daglb KD mice at 12 months of age. Scale bar: 20µm.

-In lines 380-381 the authors write: "-- Daglb, Parkin, Dj-1, and Pink1 germline KO mice all failed to develop any PD-like behavioral and pathological phenotypes."

However, this statement is not entirely true, as aged (120-week old) Parkin mutant mice do display defects in their DA neurons (Noda et al., Neurobiology of Disease, 2020), and the wording of that sentence should be adjusted accordingly.

RESPONSE: We agree with the reviewer that some of those germline KO mice, like the Parkin KO mice, did develop some modest parkinsonian phenotypes at much older age. On Page 19 of the revised manuscript, we cited the Parkin reference and rewrote the sentence as "Although some modest loss of nigral DANs and other neuropathological and motor behavioral abnormalities were observed in the aged (> 2-year-old) *Parkin* germline KO mice⁸, *Daglb* germline KO mice failed to develop any apparent neuropathological and motor behavioral phenotypes at 20 months of age."

References:

- 1 Tervo, D. G. *et al.* A Designer AAV Variant Permits Efficient Retrograde Access to Projection Neurons. *Neuron* **92**, 372-382, doi:10.1016/j.neuron.2016.09.021 (2016).
- 2 Backman, C. M. *et al.* Characterization of a mouse strain expressing Cre recombinase from the 3' untranslated region of the dopamine transporter locus. *Genesis* **44**, 383-390, doi:10.1002/dvg.20228 (2006).
- 3 Costa, R. M., Cohen, D. & Nicolelis, M. A. Differential corticostriatal plasticity during fast and slow motor skill learning in mice. *Curr Biol* **14**, 1124-1134, doi:10.1016/j.cub.2004.06.053 (2004).
- 4 Farrell, J. S. *et al.* In vivo endocannabinoid dynamics at the timescale of physiological and pathological neural activity. *Neuron* **109**, 2398-2403 e2394, doi:10.1016/j.neuron.2021.05.026 (2021).
- 5 Blauwendraat, C., Nalls, M. A. & Singleton, A. B. The genetic architecture of Parkinson's disease. *Lancet Neurol* **19**, 170-178, doi:10.1016/S1474-4422(19)30287-X (2020).
- 6 Deng, H., Wang, P. & Jankovic, J. The genetics of Parkinson disease. *Ageing Res Rev* **42**, 72-85, doi:10.1016/j.arr.2017.12.007 (2018).
- 7 Zhao, Y. *et al.* The role of genetics in Parkinson's disease: a large cohort study in Chinese mainland population. *Brain* **143**, 2220-2234, doi:10.1093/brain/awaa167 (2020).
- 8 Noda, S. *et al.* Loss of Parkin contributes to mitochondrial turnover and dopaminergic neuronal loss in aged mice. *Neurobiol Dis* **136**, 104717, doi:10.1016/j.nbd.2019.104717 (2020).

Reviewers' Comments:

Reviewer #1:

Remarks to the Author:

I thank the authors for addressing my concerns. Indeed, I had misunderstood Fig. 2; I hope that, at minimum, my confusion helped present/describe the data in a way that may be clearer to other readers. I also thank the authors for addressing my other concerns/questions thoroughly. I do not have additional reservations, and believe the manuscript to be much improved. I believe this to be an exciting story of interest to a broad neuroscience audience interested in both the biology of endocannabinoid signaling in the basal ganglia, and in the pathophysiology of Parkinson's disease.

Reviewer #2:

Remarks to the Author:

I have no additional comments to the authors.

Reviewer #3:

Remarks to the Author:

The authors have responded to the comments and made adequate revisions. I have no further remarks.

Reviewer #4:

None